# Doc-to-LoRA: Learning to Instantly Internalize Contexts

Rujikorn Charakorn [1]   Edoardo Cetin [1]   Shinnosuke Uesaka [* 2]   Robert T. Lange [1]

## Abstract

Long input sequences are central to in-context learning, document understanding, and multi-step reasoning in Large Language Models (LLMs). However, the quadratic attention cost of Transformers makes inference memory-intensive and slow. While context distillation (`CD`) can transfer information into model parameters, per-prompt distillation is impractical due to training costs and latency. To address these limitations, we propose *Doc-to-LoRA* (`D2L`), a lightweight hypernetwork that meta-learns to perform approximate `CD` within a single forward pass. Given an unseen prompt, `D2L` generates a LoRA adapter for a target LLM, enabling subsequent queries to be answered without re-consuming the original context, reducing latency and KV-cache memory consumption during target LLM inference. On a long-context needle-in-a-haystack task, `D2L` successfully learns to map contexts into adapters that store the needle information, achieving near-perfect zero-shot accuracy at sequence lengths exceeding the target LLM's native context window by more than $4\times$. On real-world QA datasets with limited compute, `D2L` outperforms standard `CD` while significantly reducing peak memory consumption and update latency. We envision that `D2L` can facilitate rapid adaptation of LLMs, opening up the possibility of frequent knowledge updates and personalized chat behavior. Code and checkpoints are available at https://github.com/SakanaAI/doc-to-lora.

## 1. Introduction

LLMs are commonly adapted to documents, tasks, and user preferences by placing relevant information in the context window, also known as prompting or in-context learning (ICL, Brown et al., 2020). While effective and convenient, ICL is transient and memory-intensive; long prompts increase latency and memory via quadratic attention and KV-cache growth. Moreover, generation quality typically degrades under longer context lengths (Liu et al., 2024; Li et al., 2025; Hong et al., 2025). A standard remedy is to compress information from context into model parameters via supervised finetuning (SFT, Zhang et al., 2023; Pareja et al., 2025). However, SFT requires collecting a task-specific dataset that represents the desired behavior and risks overfitting when data is scarce. Furthermore, repeated expensive SFT processes are required if the information, e.g., user preference, changes over time. These constraints limit the possibility of scalable model customization by practitioners.

Context distillation (`CD`) offers a compelling alternative (Askell et al., 2021; Choi et al., 2023; Padmanabhan et al., 2023; Snell et al., 2023; Bhargava et al., 2024; Caccia et al., 2025; Eyuboglu et al., 2025; Kujanpää et al., 2025; Qi et al., 2025; Shin et al., 2025). At its core, `CD` trains an LLM—without access to relevant information—to imitate its own outputs when the information is provided in context. Once the information is *internalized*[1], subsequent inference calls are significantly faster as the model does not need the information to be in its context window. Still, its training is slow and memory-intensive, making it impractical for interactive or on-device applications. Furthermore, when the information sources are constantly changing, repeated slow `CD` processes must take place.

In this work, we aim to combine the convenience of ICL with the effective internalization provided by `CD`. We propose to meta-learn the `CD` process into a *hypernetwork* (Ha et al., 2016). In other words, we meta-train the hypernetwork to represent a mapping from a given context to its corresponding weight updates produced by `CD`. Conceptually, after reading a context once, a lightweight hypernetwork generates a context-specific LoRA adapter (Hu et al., 2022). With the adapter, the target LLM can respond to subsequent queries *without* the context, reducing latency and KV-cache size. Once trained, the hypernetwork can be reused for any new context. A single, inexpensive forward pass executes

---

*S. Uesaka contributed while he was an intern at Sakana AI.
[1]Sakana AI, Tokyo, Japan [2]Minerva University, California, USA. Correspondence to: Rujikorn Charakorn <rujikorn@sakana.ai>.

*Proceedings of the $43^{rd}$ International Conference on Machine Learning*, Seoul, South Korea. PMLR 306, 2026. Copyright 2026 by the author(s).

---

[1]We define internalization concretely in Section 2. Informally, internalization is a process that transforms information into a model's parameters such that the model can later access it.

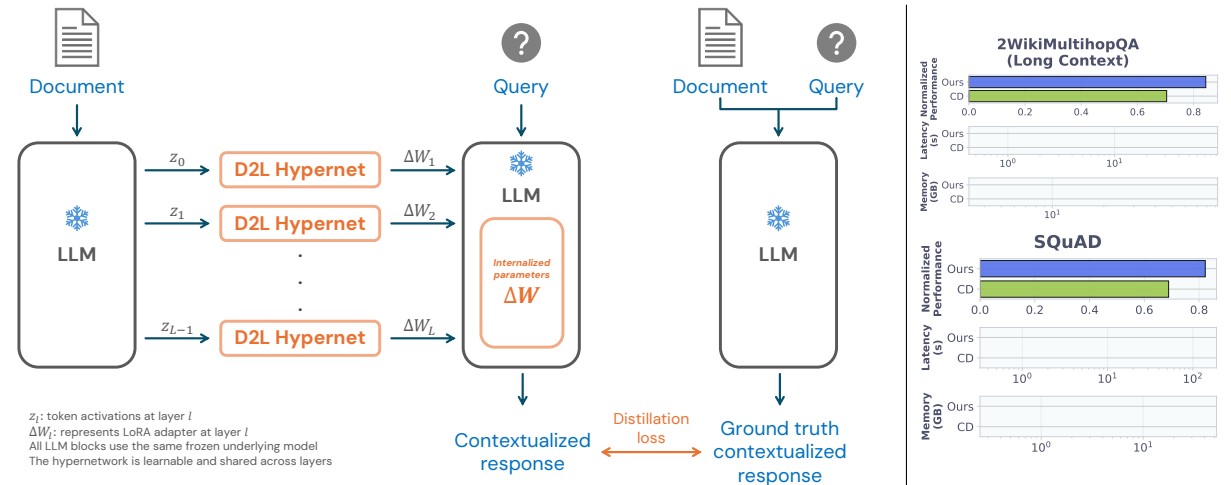

*Figure 1.* An overview of **D2L** training (left) and downstream performance (right). **D2L** learns to efficiently internalize information, outperforming traditional CD while significantly reducing latency and memory consumption across question-answering benchmarks under limited query budgets.

the learned **CD** process, enabling low-latency internalization. We call this method *Doc-to-LoRA* (**D2L**).

To enable **D2L** to internalize contexts longer than those seen during training, we base our hypernetwork on the Perceiver architecture (Jaegle et al., 2021), allowing it to map variable-length inputs to a fixed-shape LoRA adapter. Furthermore, **D2L** utilizes a chunking mechanism that allows it to produce higher-rank LoRA adapters for longer contexts without changing the output shape of the hypernetwork. This mechanism is crucial for processing documents longer than the context window of the target LLM. As a result, on a synthetic Needle-in-a-Haystack (NIAH) task, **D2L** achieves near-perfect zero-shot accuracy on contexts beyond 32K tokens, despite being trained only on sequences of up to 256 tokens, indicating that **D2L** is capable of generalizing beyond the lengths of its training data and that the chunking mechanism is effective at processing long sequences. Empirically, on real-world QA tasks, **D2L** learns an effective mapping between contexts and their corresponding LoRA adapters, outperforming **CD** under low compute budgets while significantly reducing internalization latency and compute cost. On long-document QA tasks, **D2L** generalizes in a zero-shot manner to documents exceeding the training length, without being explicitly trained to internalize such long inputs. A conceptual overview of **D2L** is shown in Figure 1. We summarize our contributions as follows:

- We introduce a meta-training objective that distills the **CD** process into a hypernetwork. This approach amortizes the overhead associated with traditional **CD** by training the hypernetwork to emulate the entire **CD** process in a single forward pass (Section 3.1).

- We introduce a well-designed architecture that makes the hypernetwork robust to varying input lengths and allows it to output higher-rank LoRAs by chunking

long inputs (Section 3.2). Consequently, **D2L** can internalize needle information with near-perfect zero-shot accuracy up to $4\times$ the maximum length of the base LLM on an NIAH task (Section 4).

- We empirically validate **D2L** and show that, under limited model update budgets, it outperforms **CD** with improved internalization efficiency, significantly reducing update latency and memory usage. Furthermore, **D2L** generalizes zero-shot to long-document QA tasks with even greater computational efficiency compared to the gains observed on shorter contexts (Section 5.1). Remarkably, **D2L** zero-shot transfers visual information effectively from a visual-language model (VLM) to a text-based LLM, allowing the target model to classify images purely through internalized information (Section 5.2). Finally, analyses of extreme generalization, performance gains, ablations, and failure modes of **D2L** are included in Section 6 and App. C.

## 2. Preliminaries

**Context Distillation** (**CD**) is a self-distillation method that internalizes behaviors induced by an in-context prompt, $c$, into a model's parameters. Unlike traditional knowledge distillation (Buciluă et al., 2006; Hinton et al., 2015), **CD** uses the same LLM for both the "teacher" and the "student." The teacher has access to the prompt $c$, while the student does not. Concretely, given a context and query pair $(c, x)$, we sample a response $y$ from the teacher, parameterized by $\theta$: $y \sim p_\theta(\cdot \mid x, c)$. Then, $y$ is used as the target for training the student with access only to the query $x$. Mathematically, the **CD** training objective can be written as

$$\min_{\theta_c} \; \Big[ \mathrm{KL}\big(p_\theta(y \mid x, c) \, \| \, p_{\theta_c}(y \mid x)\big) \Big], \qquad (1)$$

where $\mathrm{KL}(\cdot\|\cdot)$ denotes the Kullback–Leibler divergence between two probability distributions and the context-specific parameters $\theta_c$ are initialized from the original parameters $\theta$. Given that modern LLMs are effective at following instructions provided in context, CD is a practical way to internalize new knowledge or behaviors without expensive data gathering steps needed for SFT.

**Defining Internalization.** Optimizing Equation (1) risks overfitting since it only considers a single $(c, x, y)$ triplet. We refer to it as *query-dependent* distillation as the learning signal comes solely from a single query $x$. A more robust alternative is *query-independent* distillation, which utilizes multiple queries, $X = \{x_i\}_{i=1}^n$, and responses, $Y = \{y_i \sim p_\theta(\cdot|x_i, c)\}_{i=1}^n$. Collectively, $X$ and $Y$ form a small dataset $\mathcal{D}_c = \{(x_i, y_i)\}_{i=1}^n$ specifically for internalizing context $c$ with various queries and self-generated responses. Typically, $X$ is generated by prompting an LLM to generate relevant questions for the context $c$ (Caccia et al., 2025; Kujanpää et al., 2025; Eyuboglu et al., 2025). Thus, the query-independent distillation objective becomes

$$\min_{\theta_c} \;\; \mathbb{E}_{(x,y)\sim\mathcal{D}_c}\Big[\mathrm{KL}\big(p_\theta(y \mid x, c) \,\|\, p_{\theta_c}(y \mid x)\big)\Big] \quad (2)$$

We define the optimization in Equation (2) as the *internalization* process of a context $c$ into model parameters $\theta$. Successful internalization means the model can access information from $c$ through the internalized parameters $\theta_c$, behaving as if $c$ were provided in context. Context distillation has strong implications for real-world applications as it enables the creation of many specialized models without requiring explicit data collection. For instance, persistent instructions like safety and alignment prompts are prime candidates for internalization. These prompts are often tailored for different use cases and must remain in the context window throughout deployment.

## 3. Meta-Learning Context Distillation

### 3.1. Learning to Internalize Context with Hypernetworks

In this work, we focus on meta-learning query-independent CD, which allows the internalized parameters $\theta_c$ to be used with unseen downstream queries. Thus, the meta-training phase trains a hypernetwork to map a context $c$ to a set of adapter parameters that modify a frozen base model $\theta$, yielding a context-internalized model $\theta_c = \theta + \Delta W_c$, where $\Delta W_c = H_\phi(c)$ and $H_\phi$ represents a mapping from a raw string to LoRA parameters, parameterized by $\phi$ (Section 3.2). The overall meta-training process is similar to vanilla CD (Equation (2)) with one key distinction: optimizing a single $H_\phi$ that generalizes across many contexts (tasks), rather than optimizing separate $\Delta W$ per context. The hypernetwork must be exposed to a vast number of

contexts to robustly represent the CD process. Consequently, the meta-training dataset $\mathcal{D}$ must include diverse contexts, queries, and responses: $\mathcal{D} = \{c_i, \mathcal{D}_{c_i}\}_{i=1}^n$. Mathematically, we optimize $H_\phi$ to minimize the divergence between the context-conditioned teacher and the context-internalized student:

$$\min_{\phi} \;\; \mathbb{E}_{(c,\mathcal{D}_c)\sim\mathcal{D}} \; \mathbb{E}_{(x,y)\sim\mathcal{D}_c} J(x, y, c), \quad (3)$$

$$J(x, y, c) = \mathrm{KL}\big(p_\theta(y \mid x, c) \,\|\, p_{\theta+H_\phi(c)}(y \mid x)\big) \quad (4)$$

After training on a large corpus $\mathcal{D}$, the trained hypernetwork should serve as a generic mapping between context information $c$ and corresponding internalized parameters $\theta_c$, generated by a single D2L hypernetwork forward pass. That is, a trained mapping $H$ amortizes *both* the query generation process and the backpropagation needed by traditional CD. The data generation pipeline is described in App. B.1.

### 3.2. D2L Architecture

Next, we describe how D2L maps a context string into LoRA matrices (Figure 1 shows an overview of this process). A context $c$ is fed through the frozen target LLM to obtain per-layer token activations $Z \in \mathbb{R}^{L \times N \times D}$, where $L$ is the number of Transformer layers (including the embedding layer), $N$ is the number of context tokens, and $D$ is the hidden size. We denote the activations at depth $l$ by $Z_l \in \mathbb{R}^{N \times D}$, with $Z_0$ being the token embeddings. For each transformer layer $l \in \{1, \ldots, L\}$, a shared hypernetwork $h_\phi$ consumes $Z_{l-1}$ and outputs low-rank LoRA parameters (Hu et al., 2022): $h_\phi(Z_{l-1}) = \Delta W_l = B_l A_l$. Specifically, each target weight $W_l \in \mathbb{R}^{d_l^{\text{out}} \times d_l^{\text{in}}}$ is adapted as

$$W_l' \;=\; W_l \;+\; \alpha_l\, B_l A_l; \; A_l \in \mathbb{R}^{r \times d_l^{\text{in}}}, \; B_l \in \mathbb{R}^{d_l^{\text{out}} \times r}, \quad (5)$$

where $r$ is the rank and $\alpha_l$ is a learnable per-layer scalar.

**Perceiver-Based Hypernetwork:** We structure the hypernetwork $h_\phi$ as a Perceiver-style cross-attention module (Jaegle et al., 2021) that maps variable-length inputs ($Z_{l-1}$) to a fixed number of latent queries (the LoRA rank). We briefly describe the simplest configuration of the Perceiver with a single cross-attention block here.[2] Specifically, for each layer $l$, we use $r$ learnable, input-independent latent queries $Q_m \in \mathbb{R}^{r \times d_q}$. Cross-attending $Q_m$ to $Z_{l-1}$ yields $r$ latent vectors:

$$U_l \;=\; \mathrm{XAttn}\big(Q_m, K(Z_{l-1}), V(Z_{l-1})\big) \in \mathbb{R}^{r \times d_u}, \quad (6)$$

where $K(\cdot)$ and $V(\cdot)$ are linear projections. Two per-layer output heads then produce the LoRA matrices by mapping each latent to a row of $A_l$ and a column of $B_l$. Overall, the

---

[2]The cross-attention operator can be repeated many times and interleaved with latent self-attention blocks.

mapping from a context string to LoRA matrices, $H_\phi$, given context $c$ and its token activations $Z$, can be written as

$$H_\phi(c) = \Delta W = \{\Delta W_l\}_{l \in \{1,...,L\}}, \quad (7)$$
$$\Delta W_l = B_l A_l = h_\phi(Z_{l-1}) \quad (8)$$

**Long-Context Composition via Chunking:** For long contexts, we partition $c$ into $K$ contiguous chunks $\{c^{(k)}\}_{k=1}^K$ and process each chunk independently through the hypernetwork, producing per-chunk adapter $(A_l^{(k)}, B_l^{(k)})$. We then combine chunks by concatenating along the rank dimension

$$A_l = \begin{bmatrix} A_l^{(1)} \\ \vdots \\ A_l^{(K)} \end{bmatrix}, \qquad B_l = \begin{bmatrix} B_l^{(1)} & \cdots & B_l^{(K)} \end{bmatrix}, \quad (9)$$

resulting in LoRA matrices with the total rank $r \cdot K$. This composition allows **D2L** to integrate information across many chunks without changing the output shape of the hypernetwork. Note that the described architecture is not limited to generating LoRA matrices. In App. D, we empirically show that it can effectively learn to output compressed KV cache (i.e., prefix-tuning, Li & Liang, 2021).

# 4. Implanting Synthetic Needle-in-a-Haystack Information

In this section, we aim to show that **D2L** (i) successfully induces knowledge internalization, enabling the base model to recall the implanted information *without* reading the raw context, (ii) effectively bypasses the inherent context-length limitations of the base language model, and (iii) reduces the computational requirements for inference, especially when the inputs are long. For illustration purposes, we evaluate **D2L** on a synthetic needle-in-a-haystack (NIAH) information retrieval task.

Our needle-in-a-haystack (NIAH) task is primarily based on the RULER benchmark (Hsieh et al., 2024). Briefly, the NIAH task requires the model to locate a specific piece of information (needle) within a long, distracting document (haystack). The needle is a sentence defining a special 4-digit number, e.g., "The special magic number is 0042." This sentence is randomly inserted into a document filled with distractor text. The goal is to accurately retrieve the number when prompted. We use gemma-2-2b-it (Team et al., 2024) with 8K context length as the base LLM in all experiments.

During **D2L**'s meta-training, we use input contexts ranging from 32 to 256 tokens in length. The training inputs are randomly chunked from 1 to 8 chunks with a minimum chunk size of 25 tokens. We use a simplified training setup for this pedagogical experiment (see App. A for more details).

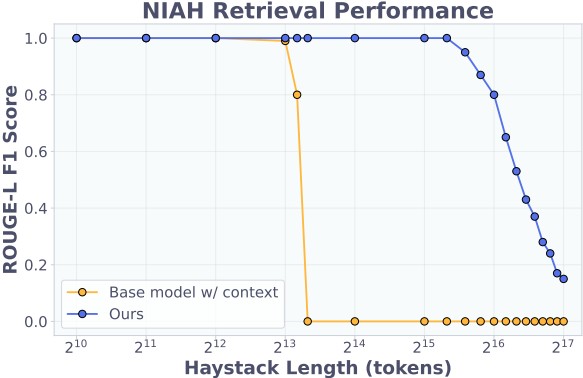

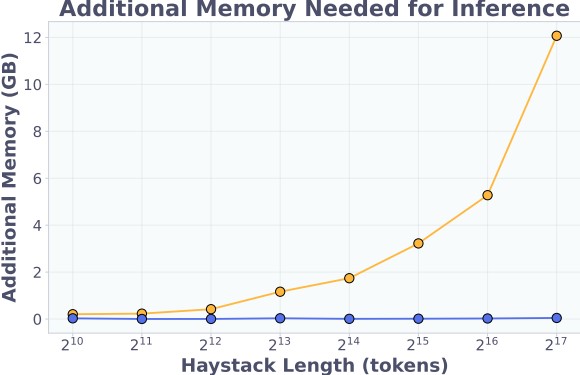

*Figure 2.* NIAH retrieval performance (top) and additional memory needed for inference (bottom).

During evaluation, the baseline has direct access to both the haystack and the query. For **D2L**, the base LLM does not have direct access to any part of the original context but is simply given the query prompt: "What is the special magic number? Reply with only the number." To perform well at this task, **D2L** must learn to map context information into a LoRA adapter that stores the value of the needle. With successful internalization, the adapted base model would be able to give a correct response based purely on the knowledge contained in the LoRA adapter. **D2L** processes the inputs by segmenting them into equal-sized chunks with 1024 tokens as the maximum chunk size. This chunk size is four times larger than the 256-token maximum sequence length seen during training.

**D2L Learns Effective CD for NIAH Task and Generalizes Beyond Base Model's Context Length:** The result in Figure 2 (left) confirms the effectiveness of our approach. **D2L** successfully internalizes the needle information and achieves perfect accuracy similar to the base model with in-context information up to 8K tokens. When the haystack is longer than 8K tokens, the base model's performance drops sharply due to its limited context length. In contrast, **D2L** maintains high retrieval accuracy across these longer sequences. Notably, performance remains close to perfect up to 40 chunks (40K tokens), which is *quintuple* the num-

ber of chunks the model has been exposed to during its training phase. Beyond this point, performance begins to degrade gracefully. These results demonstrate that `D2L` exhibits strong generalization capabilities for both chunk size and the total number of chunks.

`D2L` **Reduces Inference Cost Under Long Inputs:** The result in Figure 2 (right) shows that `D2L` not only achieves high accuracy but also demonstrates significant efficiency improvements, requiring less memory than the base model, particularly at extended context lengths. The base model uses more than 12 GB of additional memory to generate a response to a 128K-token haystack. In contrast, the model with internalized knowledge consistently uses significantly lower memory ($< 50$ MB) regardless of the length of the haystack. This result highlights a potential real-world application, where users first internalize long private documents, thereby avoiding memory-intensive KV-cache usage at inference. We include additional analyses on the chunk sizes and haystack lengths in App. C.4.

## 5. Experiments

In this section, we move from the synthetic NIAH task to a more realistic setting, where `D2L` has to learn to approximate generic `CD` from a large corpus of English documents. We evaluate the ability of `D2L` to operate as a context internalizer for question-answering tasks on 6 real-world benchmarks, including short-passage and long-document QA tasks. A key advantage of `D2L` on these tasks is its ability to provide instant and inexpensive internalization, which allows the base LLM to answer subsequent queries without consuming the context window.

**Experimental Setup:** We meta-train `D2L` on contexts from a 900M-token subset of FineWeb-Edu (Lozhkov et al., 2024) and passage-grounded QA datasets including PwC (Chevalier et al., 2023), SQuAD (Rajpurkar et al., 2016), ROPES (Lin et al., 2019), and DROP (Dua et al., 2019), yielding approximately 3.2M unique contexts after filtering. For FineWeb-Edu contexts, we generate 10 grounded queries per sample with `gemma-3-12b-it` (Team et al., 2025); for each context–query pair, we sample a response and top-16 token logits from the `gemma-2-2b-it` base model as meta-training targets. See App. B.1 for the full data generation pipeline. Our Perceiver-based `D2L` has 8 cross-attention blocks without self-attention layers. It splits the inputs into equal-sized chunks with 8K tokens and outputs a rank-8 LoRA adapter for each context chunk. Each generated adapter is applied to the "down projection" layer of each MLP block of the base model. In total, it has only 309M trainable parameters. During evaluation, `D2L` can operate in batched and iterative LoRA generation. In batched mode, it batches the token activations $Z_l$ across all layers, producing LoRA adapters for all the layers within a single

forward pass. Iterative mode, on the other hand, produces the adapter one layer at a time. The two modes allow us to prioritize either speed (batched) or lower memory consumption (iterative).[3]

We consider the following *in-parameter knowledge* baselines—where the model has to answer input queries based on its internal knowledge: (i) `CD` **(oracle)** serves as the empirical upper-bound of in-parameter knowledge methods as it optimizes the internalized weights directly on the target query (Equation (1)), (ii) `CD` trains the internalized weights using generated queries from `gemma-3-12b-it` (Equation (2)), (iii) **T2L** (Charakorn et al., 2025) is a hypernetwork-based method that directly maps the context to LoRA, similar to `D2L`. However, it was trained on SFT datasets with next-token prediction loss on ground-truth tokens, and (iv) **Base model (without context)** serves as the lower bound, as it does not have access to any context information. Both `CD` baselines internalize the context into a rank-8 LoRA adapter applied at the "down projection" layer. We use the publicly available checkpoint for T2L, which has been trained to output rank-8 adapters for the K and V projection layers. To better contextualize the reported performance, we also include *in-context knowledge* baselines—where the model can simply retrieve the answer directly from the provided context: (i) **Base model with context** serves as the upper-bound as it can directly access the answer in context, and (ii) **LLMLingua-2** (Pan et al., 2024), a prompt compression method. Results of other base models (`Mistral-7B-Instruct-v0.2` and `Qwen3-4B-Instruct-2507`) are in App. E.

### 5.1. Question-Answering from Implanted Knowledge

The following experiments test `D2L`'s ability to accurately store and retrieve information effectively from generated LoRA adapters. We report the test performance on unseen instances and use word-level ROUGE-L F1 scores as the main task performance metric. The reported performance is relative to the base model with direct access to the contexts. We provide its absolute performance in Table 12.

#### 5.1.1. EFFICIENT AND EFFECTIVE INTERNALIZATION ON READING COMPREHENSION TASKS

We assess performance on three standard reading comprehension benchmarks: **SQuAD** (span extraction, Rajpurkar et al., 2016), **DROP** (discrete reasoning over passages, Dua et al., 2019), and **ROPES** (reasoning with background knowledge, Lin et al., 2019). As shown in Figure 3 (left), `D2L` outperforms all the in-parameter baselines across all three benchmarks (see more results in App.

---

[3]Both modes are mathematically equivalent. Any task performance differences are caused by different low-level matmul kernels and the non-commutative nature of floating-point operations.

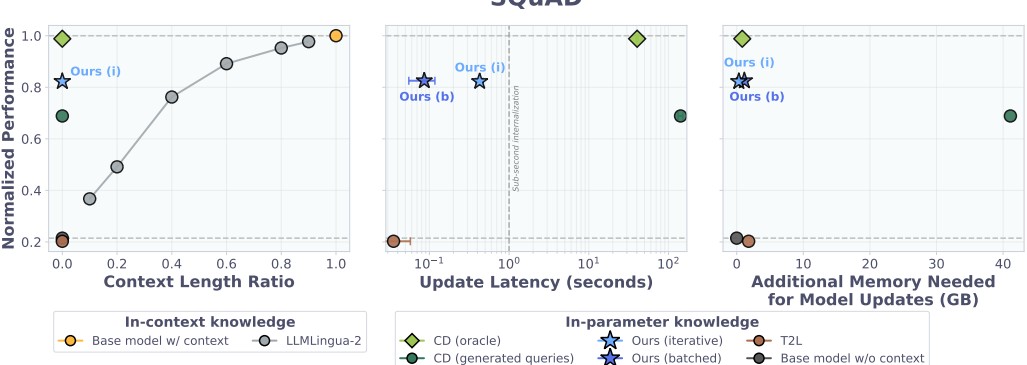

*Figure 3.* QA performance on SQuAD compared to the used context length ratio (left), update latency (middle), and additional memory needed for model updates (right). LLMLingua-2 compresses the input with [10%, 20%, 40%, 60%, 80%, 90%] compression rates from right to left (gray dots).

C), achieving $82.5\%$ relative performance compared to the ICL upper bound on the SQuAD benchmark. Compared to LLMLingua-2, **D2L** performs roughly similarly to compressing the context down to $40\%$ of its original length, but it does so while removing the context entirely.

The main benefit of using **D2L** is its update efficiency. **D2L** not only outperforms the **CD** baseline given limited compute and time constraints, it significantly reduces update latency and memory (Figure 3, middle and right). Specifically, **D2L** instantly internalizes context information within less than a second using either batched or iterative mode. **CD** (oracle) uses around $40$ seconds to internalize information due to repeated stochastic gradient descent updates. Vanilla **CD** requires even more time to internalize because of the additional query generation process, totaling more than $100$ seconds. While T2L can instantly update the model, it cannot effectively internalize knowledge due to its highly focused training data.

**D2L** requires low additional memory during the internalization process, similar to that of **CD** (oracle). Both use less than 2 GB of VRAM during the update process across the three benchmarks. In contrast, **CD** with generated queries uses more than $40$ GB due to backpropagating on many queries at once. This result highlights the main practical benefit of **D2L**: providing effective instant internalization with low memory requirement. Due to the space limit, we provide results on DROP and ROPES benchmarks in App. C. The overall trends and observations are similar to the results on the SQuAD dataset shown in Figure 3. In the next experiment, the memory efficiency of **D2L** is much more impactful when considering longer context tasks.

### 5.1.2. INSTANT ZERO-SHOT INTERNALIZATION OF LONG-CONTEXT INFORMATION

We extend our evaluation to long-context scenarios, which pose a significant challenge for standard **CD** due to memory and computational constraints of distilling long-context,

using three document-grounded QA benchmarks from Long-Bench dataset (Bai et al., 2023), **2WikiMultihopQA** (Ho et al., 2020), **MultiFieldQA**, **QASPER** (Dasigi et al., 2021), and **OOLONG** (Bertsch et al., 2025). The length of test samples can go up to $64K$ tokens. We note that **D2L** has never seen such long sequences during training. Specifically, the longest training sample is $2,344$ token long (see Figure 5). In this experiment, **CD** uses generated queries from truncated documents since the base model performs worse without truncation.

The results in Table 1 and Figure 4 show that **D2L** can effectively internalize long-context documents without being explicitly trained to do so. Like the previous experiment, **D2L** outperforms **CD** with generated queries and almost reaches the upper-bound performance of the oracle **CD** on 2WikiMultihopQA. Even with 5 queries, **CD** uses up to 79 GB of VRAM to internalize the documents. Furthermore, the oracle **CD** requires more than 7 GB of VRAM during the update. In contrast, **D2L** with iterative LoRA generation uses 2x less memory compared to the oracle's update while maintaining sub-second internalization. Results on other benchmarks are included in App. C. Once again, this result underscores the practical benefits of **D2L** for real-time and on-device applications.

In this experiment, we also investigate the additional memory used *during response generation* of the base LLM since it requires a non-negligible amount of VRAM to read long-context information. Figure 4 shows that **D2L** answers more accurately than the **CD** baseline while significantly reducing memory used during response generation. Specifically, the ICL baseline requires around 1 GB of VRAM while all in-parameter knowledge methods require less than $100$ MB.

Interestingly, on 2WikiMultihopQA and MultiFieldQA datasets, we observe that after internalizing a long context via **D2L**, the performance of the LLM slightly improves when reading the context directly again (indicated by `Ours + truncated context` in Figure 4). Also, when used

*Table 1.* Performance, update memory, and latency of in-parameter knowledge methods on 2WikiMultihopQA benchmark.

| Method | Normalized Performance ($\uparrow$) | Additional Update Memory (GB, $\downarrow$) | Mean Update Latency (s, $\downarrow$) |
|---|---|---|---|
| **CD** (oracle query) | 0.901 | 7.820 | $40.171 \pm 0.351$ |
| **D2L** (batched) | **0.857** | 11.522 | $\mathbf{0.209} \pm 0.123$ |
| **D2L** (iterative) | 0.844 | **3.791** | $0.551 \pm 0.101$ |
| **CD** (25 generated queries) | 0.745 | 59.925 | $465.454 \pm 67.868$ |
| **CD** (5 generated queries) | 0.704 | 79.371 | $72.537 \pm 7.821$ |

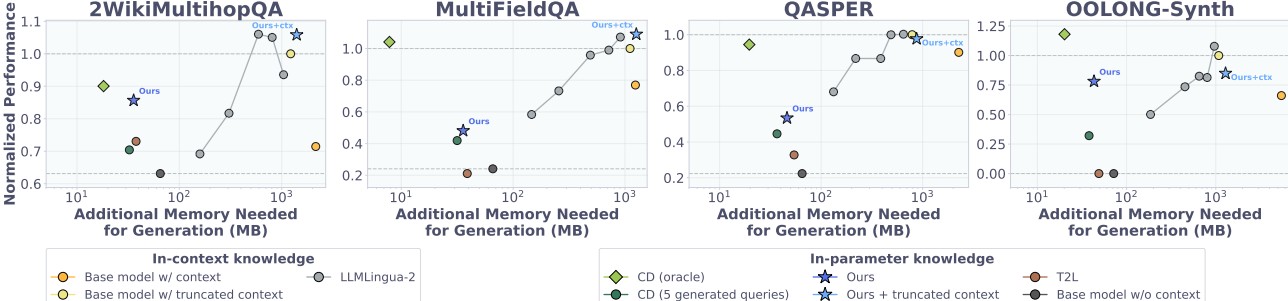

*Figure 4.* Long document QA performance. LLMLingua-2 compresses the input with $[20\%, 40\%, 60\%, 80\%, 90\%]$ compression rates from right to left (gray dots).

with LLMLingua-2 with specific compression rates, performance improves over feeding the base model uncompressed truncated inputs. We hypothesize that this might be because of the *lost-in-the-middle* (Liu et al., 2024) or *attention noise* (Ye et al., 2025) phenomena. Intuitively, text-based compression reduces unnecessary words from the context, and, therefore, can reduce attention noise and improve performance. While **D2L** produces similar improvement, we believe that the improvement comes from different mechanisms. When the model faces strong attention noise or the answers are truncated, the LLM then falls back on using its internal knowledge (Tao et al., 2024), which has been updated by **D2L**. We do not observe this effect in any of the benchmarks used in the previous experiment, presumably because attention noise does not occur under shorter context lengths.

### 5.2. Zero-Shot Internalization of Visual Information

*Table 2.* Using a VLM as the context encoder.

| | SQuAD | DROP | ROPES | Imagenette |
|---|---|---|---|---|
| **D2L** (LLM $\Rightarrow$ LLM) | **0.814** | **0.655** | **0.906** | N/A |
| **D2L** (VLM $\Rightarrow$ LLM) | 0.705 | 0.568 | 0.772 | **75.03%** |

Since **D2L** is based on the Perceiver architecture that can map arbitrary-length inputs to fixed-size outputs, we are not restricted to using the target LLM as the context encoder. In this experiment, we explore the feasibility of using the proposed architecture for bridging between a VLM (gemma-3-4b-it) and a pure-text model (gemma-2-2b-it). We want to see whether **D2L**, *without seeing any images during training*, can internalize visual information from the VLM zero-shot. Since the number of layers of the VLM and the LLM are not the

same, **D2L** maps the activations from the first 26 layers of the VLM to the corresponding layers of the target LLM. We keep the rest of the training setup identical to the main experiment. Only the language modeling part of the VLM is used during training. To test whether the target model can classify images based solely on the internalized information, we use the Imagenette dataset (Howard, 2019), a 10-class subset of ImageNet (Russakovsky et al., 2015). The target model is prompted with the following text: "What is in this image? Choose exactly one of the following classes: tench, English springer, cassette player, chainsaw, church, French horn, garbage truck, gas pump, golf ball, parachute. Respond with only the correct class without any other text." Achieving better-than-random accuracy (10%) on this task means that **D2L** can zero-shot transfer visual information to a text-only LLM.

The results are shown in Table 2. **D2L** successfully learns to map the VLM's activations into LoRA matrices of the target LLM. Although using a VLM as the context encoder negatively impacts text-based QA performance, it enables us to directly communicate visual information extracted by the VLM to the parameters of the target text-only model. Specifically, the target model achieves 75.03% accuracy purely through the internalized information. This is a striking result given that **D2L** and the target LLM have never seen information from any other modalities except text. **D2L** training curves, integration with RAG, knowledge interference, and chunk size analyses can be found in App. C.

*Table 3.* SQuAD query internalization.

| Method | Recall | Precision |
|---|---|---|
| Base model w/ context | 0.886 | 0.876 |
| D2L | 0.740 | 0.720 |
| D2L (swapped) | 0.587 | 0.044 |
| Base model w/o context | 0.185 | 0.205 |

## 6. Analyses

### 6.1. Zero-Shot Query Internalization

To further test the robustness of D2L, we use the existing SQuAD dataset with a slight modification. Instead of internalizing the document, we use D2L to internalize the query. This means that, during evaluation, the target model sees the document directly but not the query. This experiment represents an extreme generalization test for D2L, as it is trained to internalize knowledge, not queries. The results in Table 3 are encouraging and suggest that D2L can function properly under such an extreme generalization test. In this "swapped" configuration, while D2L performs worse, it still achieves moderate performance on the ROUGE-L recall metric (0.587) and outperforms the no-context baseline (0.185). We observe that the query-internalized model occasionally generates correct answers, but its outputs are typically verbose, leading to a sharp decrease in precision. This test suggests that D2L could be used to internalize other kinds of information beyond factual information from documents. Qualitative examples of this behavior are provided in Figure 13.

### 6.2. D2L Emulates CD over Many Generated Queries

*Table 4.* SQuAD (100 samples) performance with varying queries.

| Method | Normalized Performance | Update Latency (s) |
|---|---|---|
| CD (oracle) | 0.988 | $8.763 \pm 0.313$ |
| D2L | **0.866** | **0.086** $\pm$ 0.061 |
| CD (100 generated queries) | 0.650 | $631.101 \pm 12.879$ |
| CD (50 generated queries) | 0.601 | $311.661 \pm 7.211$ |
| CD (20 generated queries) | 0.506 | $129.044 \pm 5.813$ |

In the main experiments (Section 5), D2L consistently outperforms vanilla CD under small query budgets. We hypothesize that D2L performs better because the hypernetwork learns to emulate the effect of distilling over a large number of queries per context. Although each training example provides only 10 generated queries, training across millions of context samples exposes the hypernetwork to a larger and more diverse query distribution. In other words, the training samples collectively regularize D2L to be robust to a wider variety of queries than presented in any single sample.

Empirically, increasing the number of generated queries for vanilla CD improves its performance, but it remains well below that of D2L (Table 4). On a 100-sample subset of SQuAD, its performance rises from 0.506 (20 queries) to 0.650 (100 queries). Crucially, CD with 100 queries takes more than 10 minutes to internalize *each sample*. In contrast, D2L achieves 0.866, substantially closer to the oracle CD upper bound, without incurring per-sample query generation or backpropagation costs. Although increasing the budget to hundreds or thousands of queries would likely boost CD performance beyond that of D2L, the update latency would also increase substantially. Such a large latency would render the user experience non-responsive, especially in fast-changing environments, e.g., active codebases or agentic tasks. This result highlights the effectiveness of D2L at instantly internalizing knowledge in the sub-second regime. This trend supports the interpretation that the hypernetwork has learned a mapping that approximates the CD process with many more queries per context. A complementary explanation is that D2L has learned a specialized but biased form of CD. Specifically, D2L might assume that subsequent queries will always be related to the internalized knowledge (Table 8).

### 6.3. Data Ablation

*Table 5.* Data ablation

| | SQuAD | DROP | ROPES |
|---|---|---|---|
| D2L | 0.814 | **0.655** | 0.906 |
| D2L w/o QA data | **0.838** | 0.574 | **0.923** |
| CD (20 queries) | 0.689 | 0.504 | 0.776 |

In previous experiments, training samples from QA tasks (SQuAD, DROP, ROPES) are included in training to ensure that D2L learns to respond correctly to the question formats presented in those datasets. Here, we remove samples from the QA tasks and exclusively use the training data derived solely from the pretraining FineWeb-Edu corpus. Table 5 shows that the overall performance of D2L without QA data is comparable to the default training setup. Notably, it slightly outperforms the default D2L on SQuAD and ROPES, while underperforming significantly on DROP due to differences in training data distributions. Regardless of the QA data, D2L consistently outperforms low-budget CD (20 queries) across datasets, suggesting that D2L is largely robust and not highly sensitive to the training data format.

### 6.4. Ablating Training Objective

*Table 6.* Training loss and data ablation.

| Method | SQuAD (normalized F1) | SQuAD (swapped, recall) |
|---|---|---|
| D2L (50%, KL) | **0.819** | **0.385** |
| D2L (50%, NTP) | 0.763 | 0.235 |

This experiment compares an alternative training loss used to train the hypernetwork. By default, **D2L** minimizes the context-conditioned KL objective in Equation (4) using self-generated responses from the base LLM as distillation targets (self-responses). We compare this default approach against an alternative that replaces KL with the next-token prediction (NTP) loss. Due to the high cost of training the hypernetwork, we use an early checkpoint (50%) for each method in this experiment. As shown in Table 6, the KL variant outperforms the NTP variant on SQuAD, achieving a normalized F1 score of 0.819 compared to 0.763. The gap is even wider when considering the "swapped" configuration—an extreme generalization case—where the KL approach achieves a 0.385 recall score while the NTP variant only reaches 0.235. A plausible explanation is that KL distillation transfers richer information by matching the full teacher distribution $p_\theta(\cdot|x, c)$, preserving the uncertainty and alternative modes of the base LLM. Therefore, KL could propagate knowledge more effectively than NTP. This finding aligns with prior work (Padmanabhan et al., 2023; Eyuboglu et al., 2025; Caccia et al., 2025).

### 6.5. Increasing LoRA Rank

*Table 7.* LoRA rank ablation

|  | SQuAD | DROP | ROPES |
|---|---|---|---|
| **D2L** (rank-8) | 0.814 | 0.655 | **0.906** |
| **D2L** (rank-16) | **0.896** | **0.711** | 0.895 |

Effectively utilizing more training compute is important for improving the performance of modern machine learning models. In this experiment, we train D2L with more compute by increasing the rank of the generated LoRA to 16 (from 8) to probe its scalability. Table 7 shows that D2L benefits from increased training compute at a higher rank (16). We observe that rank-16 LoRA outperforms rank-8 LoRA by significant margins on two benchmarks (SQuAD and DROP). Both perform similarly on the ROPES benchmark. These results suggest that increasing the capacity of LoRA can enhance performance and demonstrate that D2L can benefit from higher-capacity parameterizations. We opt to use rank-8 LoRA for the main experiments to simplify experimentation (less VRAM and faster training).

## 7. Related Work

Hypernetworks have been used to adapt LLMs on-the-fly for different use cases (Ivison & Peters, 2022; Ivison et al., 2023; Phang et al., 2023; Lv et al., 2024; Charakorn et al., 2025; Sun et al., 2025; Liang et al., 2026; Wang et al., 2026). Notably, MEND (Li et al., 2024) trains a hypernetwork via the **CD** objective to compress few-shot examples into prefix tokens (Li & Liang, 2021). Similarly, Gisting (Mu et al., 2024) uses the **CD** objective to train "gist" tokens that compress task instructions. Cartridges (Eyuboglu et al., 2025)

aims to utilize abundant *sleep-time* compute budget for **CD** based on the prefix-tuning parameterization. In contrast to prior works, **D2L** aims to capture a generic **CD** process that can be applied to arbitrary information presented as context while being time- and memory-efficient.

A closely related method, Generative Adapter (GA, Chen et al., 2025), optimizes a hypernetwork using the next-token prediction loss on ground-truth tokens. GA first trains its hypernetwork on a pre-training corpus and then finetunes it on curated queries and responses from existing SFT datasets. In contrast, **D2L** meta-trains the hypernetwork with the **CD** objective—using primarily generated queries and self-responses—to output a context-specific LoRA adapter. Empirical results in App. 6.4 also suggest that using the **CD** objective is crucial for robust generalization of the hypernetwork. We also note that GA achieves a significantly higher ROUGE-L F1 score than the base model (see Table 13) because of its training method that directly trains the model to output the ground truth tokens via the SFT loss. However, it achieves much lower ROUGE-L recall, indicating that the improved F1 score comes solely from the fact that Generative Adapter outputs much shorter responses despite answering with less factual accuracy (represented by the ROUGE-L recall score). Finally, the use of generated queries and self-responses allows straightforward extensions of **D2L** to domains where finetuning datasets are not available.

**Hypernetworks for Meta-Learning:** Beyond LLMs, hypernetworks have a long history as meta-learners that amortize task adaptation by directly predicting parameters (von Oswald et al., 2020; Zhao et al., 2020). These methods bypass inner-loop gradient descent at test time by learning a mapping from task embedding to model weights. **D2L** adopts this principle for learning approximate **CD** for LLMs: the hypernetwork maps context information directly to parameter deltas, bypassing the overheads associated with **CD**.

**Prompt Compression:** A line of work compresses context tokens into condensed soft tokens, reducing the inference cost of LLMs on long sequences (Mu et al., 2024; Pan et al., 2024; Chevalier et al., 2023; Zhang et al., 2025a). These techniques operate directly in the token space (e.g., prompts) by reducing the number of tokens while maintaining performance. **D2L** differs by operating in parameter space via a hypernetwork that predicts weight deltas, which yields persistent, re-usable adaptations.

## Impact Statement

This paper presents work whose goal is to advance the field of Machine Learning. There are many potential societal consequences of our work, none of which we feel must be

specifically highlighted here.

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

## A. NIAH experiment details

Here, we provide more details of the NIAH experiment. We use the cross-entropy loss on the ground-truth tokens instead of the self-distillation objective for ease of experimentation. Additionally, we use a simplified architecture where `D2L` maps the token activations from the 6th layer of `gemma-2-2b-it` to LoRA of all layers. The per-layer output heads in this architecture share the same inputs. The query used for training is the same as the evaluation query. Each training input is randomly chunked with the following probabilities: 50% for 1 chunk, 12% for 2 chunks, 37.5% for 3-8 chunks with equal chances. The training samples are 32 to 256 tokens long. There is a total of 640K training samples. `D2L` is trained for 1 epoch with a learning rate $4 \times 10^{-5}$. The meta-training takes around 3 hours on a single H200 GPU. All evaluations and measurements are done with a single H200 GPU.

## B. Main Experiments Details

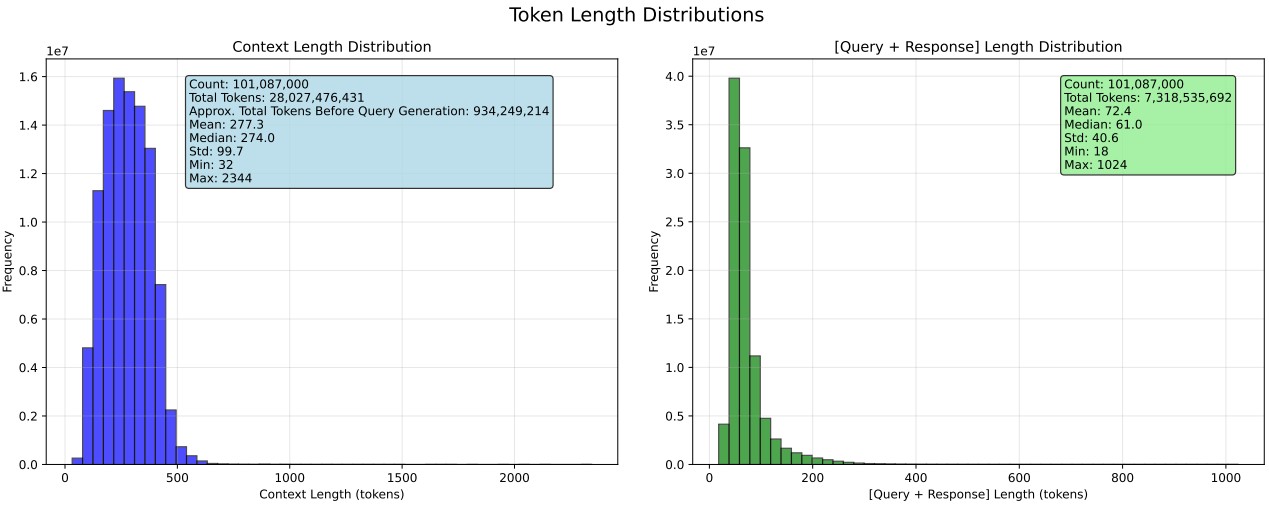

*Figure 5.* Training data length distribution. `D2L` takes contexts while the base model takes both queries and responses to compute the loss during meta-training. The total count represents the total number of unique context-query-response triplets. The number of tokens from the original contexts before query generation is roughly around 900M tokens.

### B.1. Meta-Training Data Generation Pipeline

We construct the meta-training dataset from a subset of FineWeb-Edu (Lozhkov et al., 2024), treating each sample as a context $c$. The subset contains approximately 900 million tokens. We further include passage-grounded QA datasets—PwC (Chevalier et al., 2023), SQuAD (Rajpurkar et al., 2016), ROPES (Lin et al., 2019), and DROP (Dua et al., 2019). After filtering out passages longer than $10,000$ characters, the combined corpus comprises approximately 3.2 million unique contexts. For FineWeb-Edu contexts, we generate 10 context-grounded queries per sample using `gemma-3-12b-it` (Team et al., 2025). We use a larger model for generating queries, ensuring better quality control and that the generated queries are mostly grounded in context. In practice, one could simply use the base model for generating queries. We prompt the model in two iterations, producing five queries per iteration. The first iteration uses Listing 4. The second uses Listing 5, which includes all previously generated query–answer pairs as in-context examples to encourage non-overlapping and increasingly challenging queries. The generated answers are discarded and not used for training. We augment the generated queries by putting each sample into a template instruction randomly chosen from Listing 6. We do the same for other datasets except PwC. For each unique context–query pair, we sample a single response from `gemma-2-2b-it` using Listing 7 and record the top-16 token logit values for every generated token. The logit values are the training target used in Equation (4). The overall training data length distribution is shown in Figure 5.

### B.2. Training and Experimental Details

All evaluations and measurements are done with a single H200 GPU. We set the maximum number of test samples in each dataset to be 500 due to the high overhead latency of `CD`. We use the same query template for all the benchmarks except for

QASPER (see Listing 8).

The hypernetwork consists of two modules: a Perceiver-style cross-attention encoder that consumes per-layer token activations (Listing 1) and output heads that map the latent queries to LoRA matrices (Listing 2). A pseudocode for the forward pass of the internalization process is shown in Listing 3.

We observe that the training can be unstable if `D2L` is trained to output multiple LoRAs from the beginning. Thus, we train `D2L` in a two-stage learning setup. First, `D2L` is trained to always output only one chunk for each input context for 80K gradient steps. This stage trains `D2L` to purely internalize information without emphasis on the compositionality of the generated LoRAs. Then, similar to the NIAH experiment, we randomly chunk each input with the following chunking probabilities: 50% for 1 chunk, 12% for 2 chunks, 37.5% for 3-8 chunks with equal chances. `D2L` is trained for 20K steps under the chunking setup. This stage regularizes `D2L` to output composable LoRAs. Both stages use the same data distribution shown in Figure 5. During training, each batch packs the context inputs into a 4K-token sequence and uses gradient accumulation, totaling more than 200K context tokens across 8 GPUs. We observe that this is crucial for good performance. Otherwise, the training converges too early with smaller batch sizes.

## C. Additional Results (Gemma-2-2B)

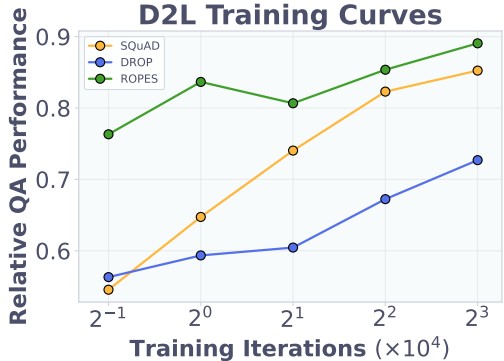

*Figure 6.* Relative QA performance from various checkpoints throughout the training process.

### C.1. Knowledge Interference

We investigate further whether `D2L` can cause *knowledge interference*, where the internalized knowledge overrides existing internal knowledge in the base LLM. `D2L` is not trained to retain existing knowledge of the target model. Thus, the goal of this experiment is to observe the behavior of `D2L` without a dedicated knowledge preservation mechanism and set a baseline for how our simple training pipeline induces knowledge interference. To study this phenomenon, we replace the context of each sample in the SQuAD dataset with either an *assistant* prompt or a *distracting* prompt. The assistant prompt is simply the

*Table 8.* SQuAD with replaced contexts.

| Method | SQuAD (assistant) | SQuAD (distracting) |
|---|---|---|
| Base model w/ replaced context | 0.201 | 0.175 |
| `D2L` | 0.096 | 0.126 |
| `CD` (10 generated queries) | 0.211 | 0.203 |

text ``You are a useful assistant.`` while the distracting prompt is a book chapter from Project Gutenberg (∼ 4K tokens). This dataset allows us to test `D2L` when the internalized knowledge and the queries are unrelated. Table 8 shows that `D2L` significantly reduces the performance compared to the base model and `CD`. We hypothesize that `D2L` might acquire a strong prior, assuming that the subsequent queries will always be related to the internalized knowledge. This might be caused by the bias in the data generation pipeline that only considers queries related to the input context. We think that by adding irrelevant queries into the training data, one could regularize the hypernetwork and help mitigate this bias. Other dedicated continual learning mechanisms, such as constraint edits (Fang et al., 2025) and sparse memory finetuning (Lin et al., 2025), could be incorporated to directly tackle the knowledge interference problem in future work.

*Table 9.* Integration of `D2L` with BM25-based RAG on long-context QA benchmarks. Scores are normalized by the base model with the full in-context document. Top-$k$ denotes the number of retrieved chunks provided in the prompt. Improvements in parentheses compare RAG + `D2L` against the corresponding RAG baseline.

| Method | 2WikiMQA | | | MultiFieldQA-EN | | | Qasper | | | Oolong-Synth | | |
|---|---|---|---|---|---|---|---|---|---|---|---|---|
| | No context | Top-1 | Top-4 | No context | Top-1 | Top-4 | No context | Top-1 | Top-4 | No context | Top-1 | Top-4 |
| D2L | **0.859** | N/A | N/A | **0.519** | N/A | N/A | **0.497** | N/A | N/A | **0.781** | N/A | N/A |
| RAG | N/A | 0.368 | 0.863 | N/A | 0.695 | **1.144** | N/A | 0.429 | 0.640 | N/A | 0.679 | 1.079 |
| RAG + D2L | N/A | **0.877** (↑0.509) | **0.966** (↑0.103) | N/A | **0.815** (↑0.120) | 1.090 (↓0.054) | N/A | **0.599** (↑0.170) | **0.776** (↑0.136) | N/A | **1.176** (↑0.497) | **1.129** (↑0.050) |

## C.2. Retrieval Augmented Generation (RAG) Integration

`D2L` is complementary to retrieval-augmented generation (RAG): RAG selects a small number of passages to expose explicitly in the prompt, whereas `D2L` internalizes the full document into the model parameters. We therefore evaluate whether the two mechanisms can be combined at inference time. In the RAG baseline, BM25 retrieves either the top-1 or top-4 chunks from the document and the base model answers using only those retrieved chunks. In the hybrid setting, `D2L` first internalizes the full document, and the same retrieved chunks are additionally prepended to the question prompt. This tests whether explicit retrieval can provide localized evidence while `D2L` supplies a broader document-level memory, especially when the answer depends on information not perfectly captured by lexical retrieval.

As shown in Table 9, combining RAG with `D2L` generally improves over retrieval alone. The gains are largest when retrieval is sparse: with only the top-1 retrieved chunk, RAG + `D2L` improves over RAG by 0.509 on 2WikiMultihopQA, 0.120 on MultiFieldQA-EN, 0.170 on QASPER, and 0.497 on Oolong-Synth. This suggests that internalizing the full document helps compensate for missing or incomplete retrieved evidence, a common failure mode when a single lexical chunk does not contain all information needed for the answer. With top-4 retrieval, the RAG baseline becomes stronger, but the hybrid method still improves on three of the four benchmarks. The exception is MultiFieldQA-EN, where top-4 RAG slightly outperforms RAG + `D2L` (1.144 vs. 1.090), indicating that when retrieved evidence is already sufficient and highly relevant, the additional internalized memory can provide less benefit or introduce mild interference.

Overall, these results indicate that `D2L` should not be viewed as a replacement for RAG. Instead, it provides a complementary context pathway: RAG offers explicit, inspectable evidence in the prompt, while `D2L` gives the model access to information from the full document without increasing generation-time context length. This combination is particularly useful in low-retrieval-budget settings, where retrieval precision is imperfect but prompt length and latency must remain constrained. Implementation details for the RAG wrapper are provided below.

## C.3. Retrieval Augmented Generation (RAG)

We implement retrieval-augmented generation (RAG) as a lightweight evaluation-time wrapper over the model's standard generation interface, rather than as a separately trained retriever-generator system. For each example, the wrapper first decodes the question from the prompt input and decodes the full document context from the provided 'ctx_ids', stripping model-specific context affixes so that retrieval operates over clean text. The decoded context is then re-tokenized and partitioned into overlapping token windows. Chunks are scored against the question using a local BM25 retriever, which provides a simple lexical matching baseline without introducing an additional embedding model or external vector index. The highest-scoring chunks are selected greedily under a retrieval budget, merged when they overlap in the original document, and then inserted into a new prompt of the form: instruction, retrieved context, and question. The final prompt is passed through the model's chat template before generation.

We use two closely related variants. In the raw RAG baseline, retrieval replaces the original long context entirely: the base language model receives only the reconstructed prompt containing the retrieved evidence. In the hybrid RAG setting, retrieval is combined with Doc-to-LoRA (D2L). Here, the full document is still internalized through the D2L pathway, but the prompt-side question input is rewritten to include the retrieved evidence. This design cleanly separates two roles of context: D2L provides model-side document internalization, while RAG provides explicit prompt-side evidence selection. The implementation supports only batch size 1 for these baselines and records per-example retrieval metadata, including the selected chunks and retrieval settings, so that retrieval behavior can be inspected during analysis.

Our default configuration uses BM25 retrieval with a chunk size of 256 tokens, an overlap of 64 tokens, top-4 retrieval, and a maximum retrieved-context budget of 1536 tokens. These defaults were chosen to provide a stable and interpretable baseline that balances retrieval precision, recall, and prompt length. A chunk size of 256 tokens is small enough to localize

relevant evidence for lexical retrieval, while still large enough to preserve paragraph-level coherence. A 64-token overlap reduces boundary failures when supporting information falls near chunk edges without introducing excessive redundancy. Retrieving up to four chunks provides enough capacity for multi-span or multi-hop evidence aggregation, while avoiding the prompt noise that can arise from larger retrieval sets. Finally, the 1536-token budget keeps prompt growth controlled while remaining permissive enough that, under the default chunk geometry, the effective limit is usually governed by top-k rather than by the token cap itself. In addition, when the full context is already shorter than the retrieval budget, the wrapper simply uses the full context directly, avoiding unnecessary retrieval-induced truncation on short examples. Together, these choices make the default RAG setting a strong but simple baseline for comparing raw retrieval against hybrid retrieval-plus-internalization.

## C.4. NIAH Ablations

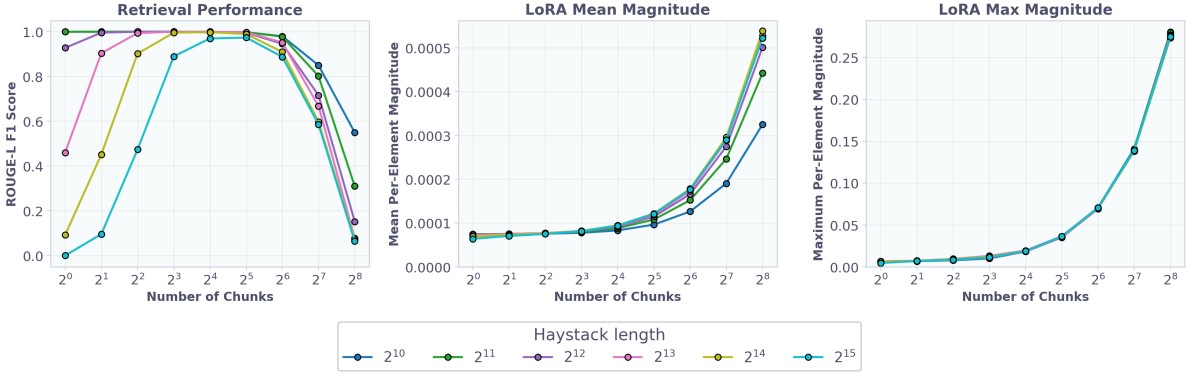

*Figure 7.* NIAH ablation varying the number of chunks (x-axis) and haystack lengths (colors). We report retrieval performance together with the mean and maximum LoRA magnitude.

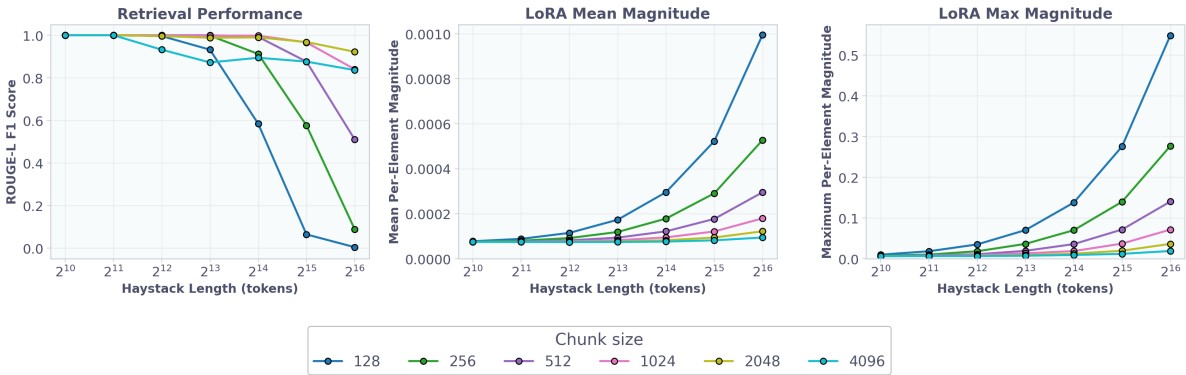

*Figure 8.* NIAH ablation varying the haystack lengths (x-axis) and evaluation chunk sizes (colors). We report retrieval performance together with the mean and maximum LoRA magnitude.

We further analyze the NIAH setting by varying the chunking configuration at evaluation time. Figures 7 and 8 show that `D2L` is largely robust to both the chunk size and the number of chunks. Still, if the chunk size is too big ($\geq 2048$ tokens) or there are too many chunks ($\geq 64$ chunks), retrieval performance starts to degrade. This is likely an artifact of our meta-training setting, where the model is trained with at most eight chunks and much shorter sequences. Our leading hypothesis is that concatenating multiple LoRAs causes noise accumulation, as shown by the increasing mean and maximum LoRA magnitudes. In the NIAH experiment, `D2L` could in principle learn to output all-zero LoRAs for non-needle chunks and output meaningful LoRAs only for chunks containing the needle information. However, because training uses only up to eight chunks, the loss never strongly regularizes this behavior over many composed chunks. Instead, `D2L` appears to learn to output LoRAs that are small enough for non-needle chunks to avoid flipping the prediction. This strategy works well up to a certain number of chunks before breaking down around 40–60 chunks. We believe that better architecture choices, such as auto-regressive LoRA generation, and regularization methods, such as filtering redundant chunk information, could help tackle this problem.

## C.5. Long-Context Benchmark Ablations

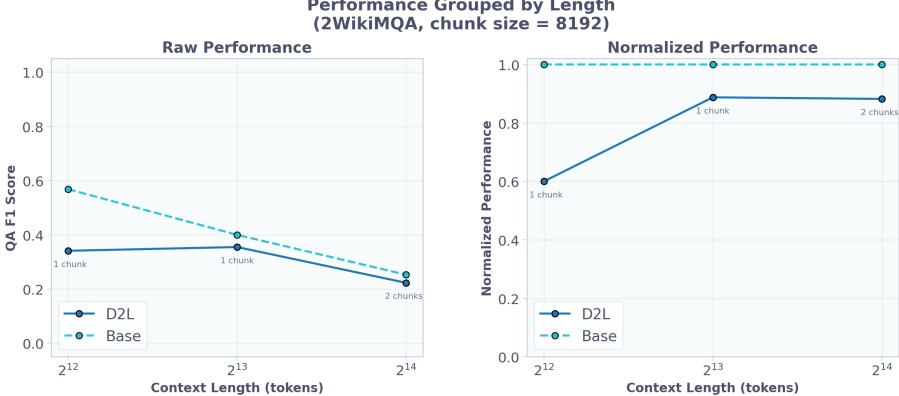

*Figure 9.* 2WikiMultihopQA performance grouped by input length when **D2L** uses an 8192-token chunk size.

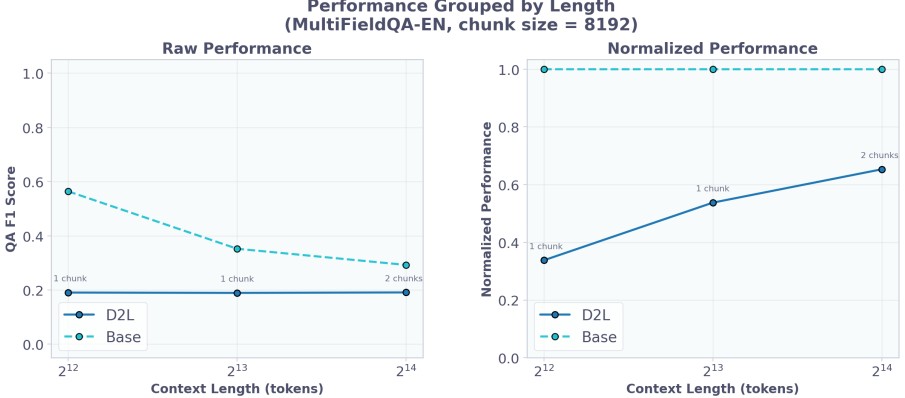

*Figure 10.* MultiFieldQA-EN performance grouped by input length when **D2L** uses an 8192-token chunk size.

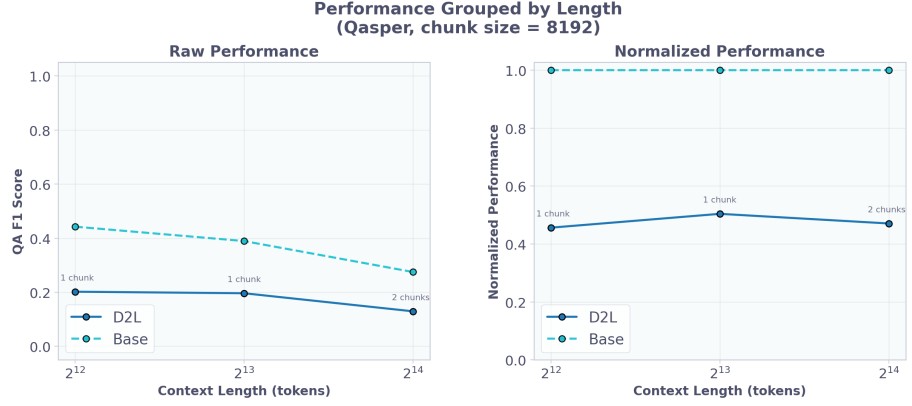

*Figure 11.* QASPER performance grouped by input length when **D2L** uses an 8192-token chunk size.

We also group long-context benchmark results by input length to test whether the gains from **D2L** are concentrated in a particular length regime. Figures 9 to 11 provide an analysis similar to the NIAH ablations by grouping long-context samples by their length. The results show that increasing the number of chunks does not negatively affect **D2L**. In fact, **D2L** degrades more slowly than the base model with full context as the context length increases. This suggests that the generated LoRAs remain composable across realistic long-context inputs, even though **D2L** was trained on much shorter sequences.

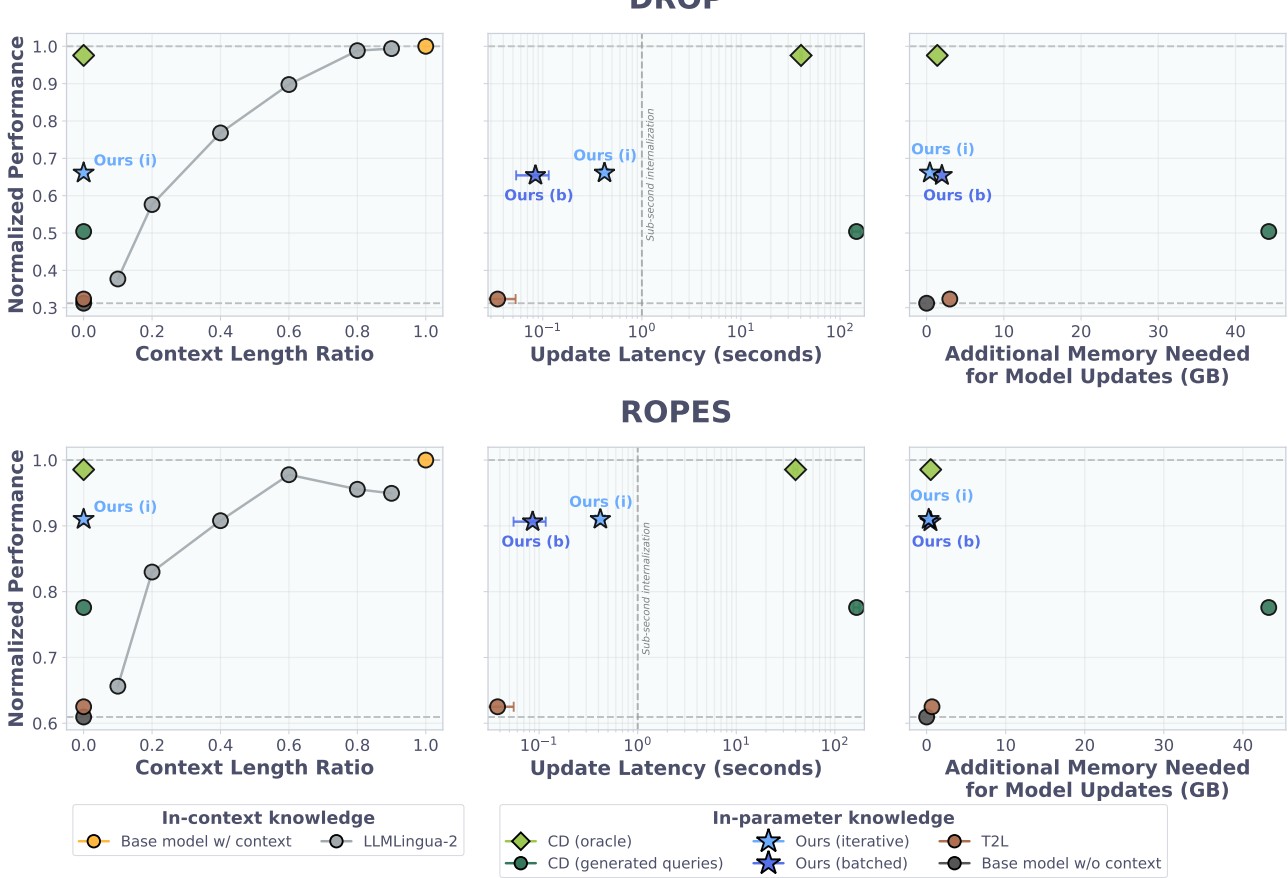

*Figure 12.* QA performance on DROP and ROPES for all methods compared to the used context length ratio (left), update latency (middle), and peak memory used by model updates (right).

*Table 10.* Performance, update memory, and latency of in-parameter knowledge methods on MultiFieldQA benchmark.

| Method | Rel. Perf vs Truncated ICL (↑) | Peak Update Memory (GB, ↓) | Mean Update Latency (s, ↓) |
|---|---|---|---|
| CD (oracle query) | 1.041 | 7.820 | $41.912 \pm 2.257$ |
| D2L (batched) | 0.481 | 19.421 | $0.198 \pm 0.094$ |
| D2L (iterative) | 0.485 | 3.675 | $0.522 \pm 0.098$ |
| CD (25 generated queries, mini-batch SGD) | 0.528 | 53.232 | $431.479 \pm 70.916$ |
| CD (5 generated queries, full-batch SGD) | 0.419 | 40.231 | $81.406 \pm 7.967$ |

*Table 11.* Performance, update memory, and latency of in-parameter knowledge methods on QASPER benchmark.

| Method | Rel. Perf vs Truncated ICL (↑) | Peak Update Memory (GB, ↓) | Mean Update Latency (s, ↓) |
|---|---|---|---|
| CD (oracle query) | 0.945 | 7.665 | $41.047 \pm 1.124$ |
| D2L (batched) | 0.534 | 31.153 | $0.179 \pm 0.105$ |
| D2L (iterative) | 0.543 | 4.781 | $0.502 \pm 0.097$ |
| CD (25 generated queries) | 0.480 | 46.383 | $410.053 \pm 47.922$ |
| CD (5 generated queries) | 0.446 | 39.234 | $87.206 \pm 5.053$ |

*Table 12.* Raw benchmark performance (ROUGE-L F1 score) of the base model with truncated ICL

| | SQuAD | DROP | ROPES | 2WikiMultihopQA | MultiFieldQA | QASPER |
|---|---|---|---|---|---|---|
| Base model w/ (truncated) context | 0.8692 | 0.4541 | 0.7457 | 0.3387 | 0.3938 | 0.3839 |

**Passage**

During the 1970s and sometimes later, Western and pro-Western governments often supported sometimes fledgling Islamists and Islamist groups that later came to be seen as dangerous enemies. Islamists were considered by Western governments bulwarks against—what were thought to be at the time—more dangerous leftist/communist/nationalist insurgents/opposition, which Islamists were correctly seen as opposing. The US spent billions of dollars to aid the mujahideen Muslim Afghanistan enemies of the Soviet Union, and non-Afghan veterans of the war returned home with their prestige, "experience, ideology, and weapons", and had considerable impact.

**Query**

Answer the following question. Output only the answer and do not output any other words.
Question: What did the non-Afghan veterans returning home have in addition to their prestige?

**Model with internalized passage**

Experience, ideology, and weapons.
(ROUGE-L F1 score = 1.0)

**Model with internalized query**

The provided text states that non-Afghan veterans returned home with their prestige, "experience," and weapons.
The text also states that the non-Afghan veterans had a considerable impact.
The text does not provide information about the ideology of the non-Afghan veterans.
(ROUGE-L F1 score = 0.20)

**Passage**

In 1979, the Soviet Union deployed its 40th Army into Afghanistan, attempting to suppress an Islamic rebellion against an allied Marxist regime in the Afghan Civil War. The conflict, pitting indigenous impoverished Muslims (mujahideen) against an anti-religious superpower, galvanized thousands of Muslims around the world to send aid and sometimes to go themselves to fight for their faith. Leading this pan-Islamic effort was Palestinian sheikh Abdullah Yusuf Azzam. While the military effectiveness of these \"Afghan Arabs\" was marginal, an estimated 16,000 to 35,000 Muslim volunteers came from around the world came to fight in Afghanistan.

**Query**

Answer the following question. Output only the answer and do not output any other words.
Question: How many Muslims came from around the world to fight in Afghanistan?

**Model with internalized passage**

16,000 to 35,000
(ROUGE-L F1 score = 1.0)

**Model with internalized query**

The provided text states that "an estimated 16,000 to 35,000 Muslim volunteers came from around the world came to fight in Afghanistan."
(ROUGE-L F1 score = 0.385)

**Passage**

The Panthers finished the regular season with a 15\u20131 record, and quarterback Cam Newton was named the NFL Most Valuable Player (MVP). They defeated the Arizona Cardinals 49\u201315 in the NFC Championship Game and advanced to their second Super Bowl appearance since the franchise was founded in 1995. The Broncos finished the regular season with a 12\u20134 record, and denied the New England Patriots a chance to defend their title from Super Bowl XLIX by defeating them 20\u201318 in the AFC Championship Game. They joined the Patriots, Dallas Cowboys, and Pittsburgh Steelers as one of four teams that have made eight appearances in the Super Bowl.

**Query**

Answer the following question. Output only the answer and do not output any other words.
Question: What team did the Panthers defeat?

**Model with internalized passage**

Arizona Cardinals
(ROUGE-L F1 score = 1.0)

**Model with internalized query**

The Panthers defeated the Cardinals.
(ROUGE-L F1 score = 0.4)

*Figure 13.* Qualitative Results on SQuAD. Green highlight represents words that overlap with the ground-truth labels. The base model with internalized queries sometimes answers correctly but the responses might be verbose, which leads to significantly lower ROUGE-L F1 scores.

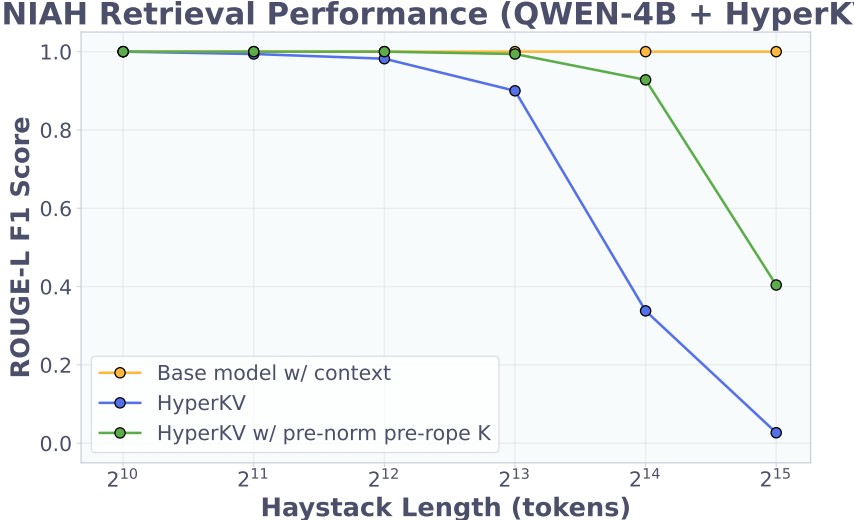

*Figure 14.* NIAH retrieval performance with generated KV cache using Qwen3-4B as the target model.

## D. Generating KV cache Instead of LoRA (`Qwen3-4B-Instruct-2507`)

This section aims to explore whether the proposed architecture can be used to generate other kinds of fine-tuning parameterizations beyond LoRA. Notably, Eyuboglu et al. (2025) show that prefix-tuning (i.e., KV caches) can be very effective at `CD`. While this parameterization does not internalize the information into the model's weights, it can still compress the KV cache significantly. The hypernetwork in this experiment outputs the equivalent of a 20-prefix-token KV cache for the target model. We use the same dataset as presented in Section 4. We note that Qwen3-4B has 128K context window, making retrieval of a single needle trivial under the haystack lengths we use for the NIAH task. Nonetheless, since the hypernetwork is trained only up to 256 tokens and 8 chunks, exhibiting good performance beyond the training range still indicates strong generalization of the hypernetwork. Similar to Section 4, the input is processed into chunks of 1024 tokens if it is longer than 1024 tokens.

**Results:** The modified hypernetwork still shows impressive retrieval accuracy beyond its training lengths. Specifically, it achieves near-perfect accuracy up to 8K tokens, beyond which performance degrades gracefully. Interestingly, we observe that the performance degrades quickly if the hypernetwork learns to generate the keys directly: The performance degradation starts around 4K tokens (Figure 14, blue line). Instead, if the generated keys are passed through the key normalization layer and the Rotary Position Embedding (RoPE, Su et al., 2024) layer, the accuracy can be sustained for much longer (Figure 14, green line). We hypothesize that, when the hypernetwork learns to directly output the keys, it has to implicitly emulate RoPE application. Because it has been trained up to 8 chunks, the performance is expected to drop sharply beyond 8 chunks, which is equivalent to 8K tokens in this experiment. The result indeed agrees with this hypothesis as the performance drops sharply beyond 8K tokens when the hypernetwork directly outputs the keys.

*Table 13.* QA performance on SQuAD with `Mistral-7B-Instruct-v0.2` as the base model. **D2L** maintains the response characteristic of the base model (similar precision and F1 scores) and high factual recall. Generative Adapter, although achieves a higher F1 score, is factually less accurate indicated by the lower recall score.

|                          | ROUGE-L Recall | ROUGE-L Precision | ROUGE-L F1 |
| ------------------------ | -------------- | ----------------- | ---------- |
| `Mistral-7B-Instruct-v0.2` | 0.919          | 0.443             | 0.519      |
| **D2L**                  | 0.835          | 0.447             | 0.515      |
| Generative Adapter       | 0.643          | 0.652             | 0.632      |

## E. Results with Other Models

To test the generality of the overall proposed method, we train **D2L** with `Mistral-7B-Instruct-v0.2` and `Qwen3-4B-Instruct-2507` as the base LLMs.

Additionally, the responses from `Mistral-7B-Instruct-v0.2` and `Qwen3-4B-Instruct-2507` are verbose despite instructing the model to output only the answer, e.g., "Answer the following question. Output only the answer and do not output any other words." This behavior makes the ROUGE-L precision and F1 scores much lower compared to `gemma-2-2b-it`. Therefore, for the `Mistral-7B-Instruct-v0.2` and `Qwen3-4B-Instruct-2507` models, we report the ROUGE-L recall score instead of ROUGE-L F1 score. Specific to `Mistral-7B-Instruct-v0.2`, we include Generative Adapter as an additional baseline using the official checkpoint provided by Chen et al. (2025). We also note that Generative Adapter achieves a significantly higher ROUGE-L F1 score than the base model (see Table 13) because of its training method that directly trains the model to output the ground truth tokens via the SFT loss. However, it achieves much lower ROUGE-L recall than that of **D2L** and the base model, indicating that the improved F1 score comes solely from the fact that Generative Adapter outputs much shorter responses despite answering with less factual accuracy (represented by the ROUGE-L recall score).

The results are presented in Figures 15 to 18. The overall finding is similar to the results collected using `gemma-2-2b-it` as the base model. **D2L** effectively internalizes new information, outperforming all the in-parameter baselines with excellent speed and memory efficiency. The results across model families and sizes indicate the robustness and generality of the proposed method.

We observe that all internalization methods struggle to update long context information for `Mistral-7B-Instruct-v0.2`. Thus, the performance of in-parameter baselines is similar to the no context baseline in this experiment. We offer no explanation as to why this LLM is harder to internalize new long-context knowledge. Investigation of this topic is beyond the scope of this work and we leave the study of such phenomenon for future work.

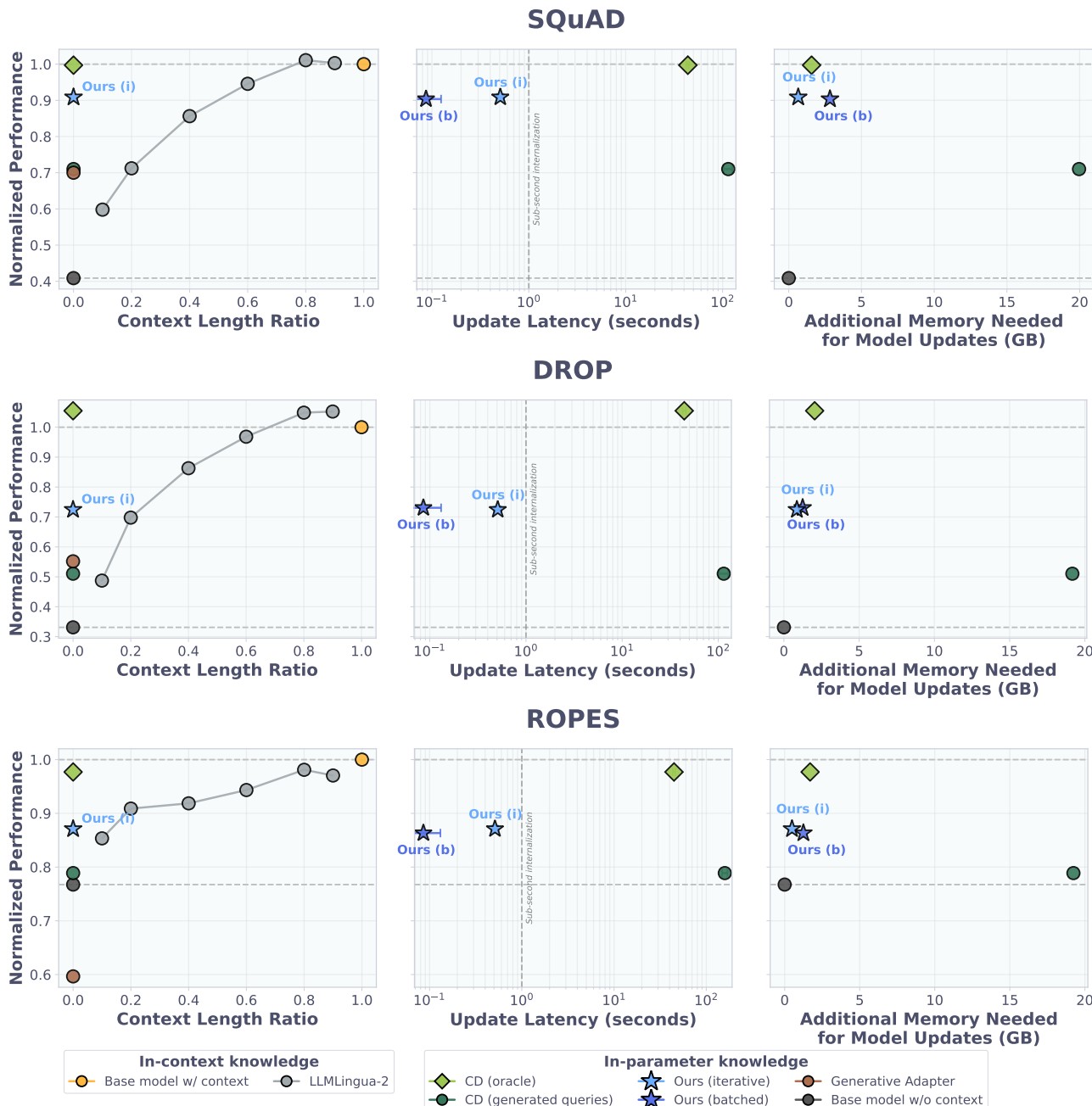

*Figure 15.* [**D2L** + `Mistral-7B-Instruct-v0.2`] QA performance on DROP and ROPES for all methods compared to the used context length ratio (left), update latency (middle), and peak memory used by model updates (right).

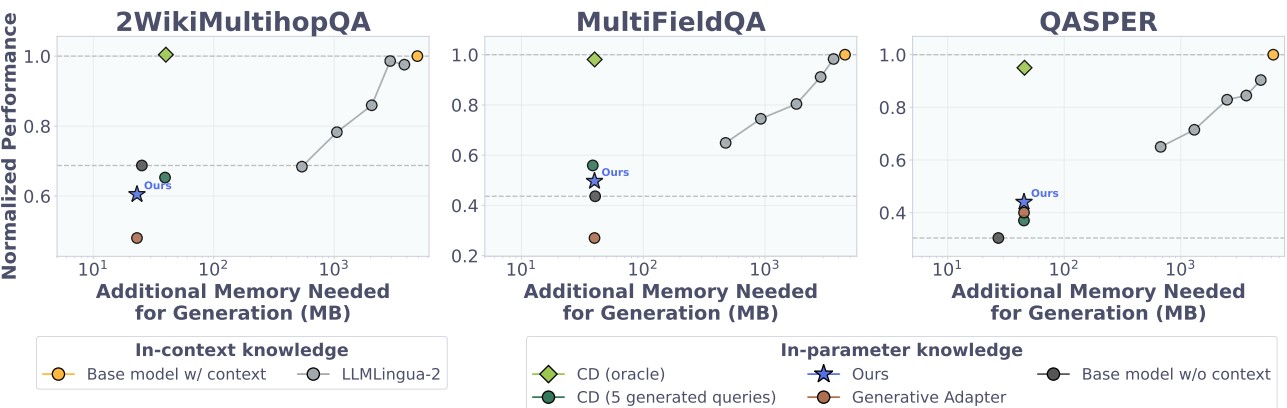

*Figure 16.* [**D2L** + `Mistral-7B-Instruct-v0.2`] Long document QA performance. LLMLingua-2 compresses the input with [20%, 40%, 60%, 80%, 90%] compression rates from right to left (gray dots).

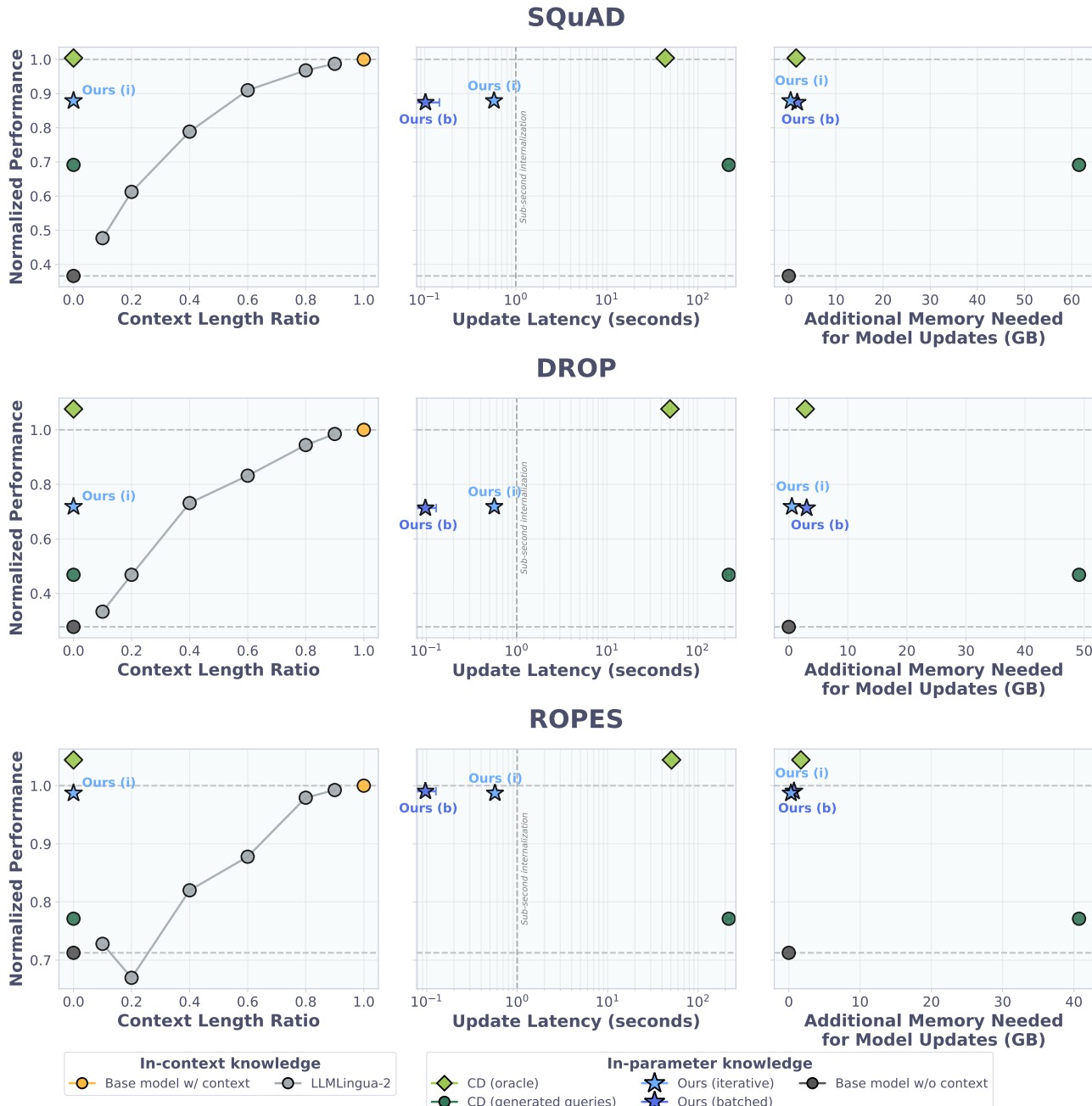

*Figure 17.* [**D2L** + `Qwen3-4B-Instruct-2507`] QA performance on DROP and ROPES for all methods compared to the used context length ratio (left), update latency (middle), and peak memory used by model updates (right).

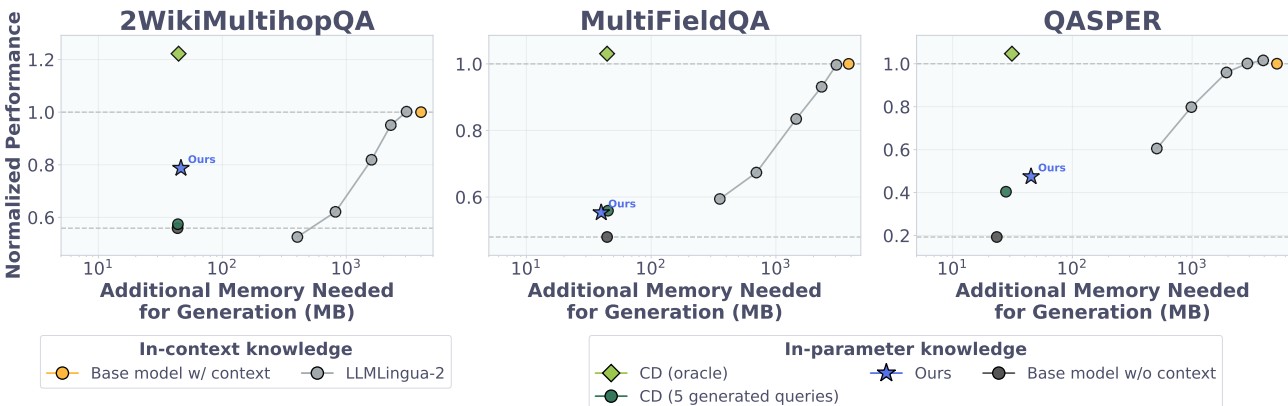

*Figure 18.* [**D2L** + `Qwen3-4B-Instruct-2507`] Long document QA performance. LLMLingua-2 compresses the input with [20%, 40%, 60%, 80%, 90%] compression rates from right to left (gray dots).

# F. Conclusion

**Summary:** We propose `D2L`, a hypernetwork that emulates the context distillation (`CD`) process. `D2L` provides instant and inexpensive knowledge internalization through a single forward pass of the hypernetwork by amortizing expensive query generation and backpropagation processes into the meta-training phase (Section 3). On a synthetic NIAH task, `D2L` successfully learns to internalize needle information and effectively extends the context window of the base model to more than $4\times$ its original length (Section 4). On real-world QA tasks, `D2L` outperforms traditional `CD`, with limited query budgets, while significantly reducing memory usage and latency of the internalization process (Section 5.1). Furthermore, we show that `D2L` can zero-shot internalize visual information from a VLM context encoder, allowing the target LLM to have basic visual understanding purely through internalized information (Section 5.2).

**Limitations:** While leading to significant cost savings compared to `CD`, `D2L` still needs a single expensive meta-training phase. Specifically, an entire training run for meta-learning `CD` for `gemma-2-2b-it` takes around 5 days on 8 H200 GPUs. Furthermore, in its current form, `D2L` requires retraining the hypernetwork for a new target LLM. We believe that significantly improving the efficiency of the meta-training phase is a fruitful research direction for future work. In general, there is a performance gap between ICL and in-parameter knowledge methods, including `D2L`. Future work could explore more performant methods for internalization, even with update overheads. One such possible approach is combining meta-learned `D2L` with `CD`, using the outputs of `D2L` as the starting point, which would be fine-tuned further by `CD`. This work focuses only on the LoRA parameterization. There are likely better parameterizations that could be more efficient, improve performance, or avoid catastrophic forgetting (Eyuboglu et al., 2025; Lin et al., 2025)

**Discussion and Future Work:** Given that `D2L` can internalize contexts with sub-second latency, we believe that `D2L` has strong implications for inference-time training techniques (Wang et al., 2024; Cao et al., 2025), which still mostly rely on backpropagation-based updates. This capability is also relevant to continual learning (Shi et al., 2025) and personalization (Zhang et al., 2025b), where the base model must iteratively incorporate new knowledge, evolving user preferences, and chat history. Furthermore, `D2L` could help shed light on machine unlearning (Bourtoule et al., 2021) and weight-space interpretability (Olah et al., 2025) by studying and manipulating internalized weights.

# G. LLM Usage Disclosure

In this paper, we use LLMs for early drafting. We also use LLMs for improving writing throughout the paper. Nonetheless, we have verified and will take full responsibility for the contents in this paper.

```
(perceiver): Idefics2Perceiver(
    (modality_projection): Idefics2MLP(
        (gate_proj): Linear(in_features=2304, out_features=9216, bias=False)
        (up_proj): Linear(in_features=2304, out_features=9216, bias=False)
        (down_proj): Linear(in_features=9216, out_features=512, bias=False)
        (act_fn): SiLU()
    )
    (encoder): Idefics2PerceiverResampler(
        (layers): ModuleList(
            (0-7): 8 x Idefics2PerceiverLayer(
                (input_latents_layernorm): Idefics2RMSNorm((512,), eps=1e-06)
                (input_context_layernorm): Idefics2RMSNorm((512,), eps=1e-06)
                (cross_attn): Idefics2PerceiverFlashAttention2(
                    (q_proj): Linear(in_features=512, out_features=2048, bias=False)
                    (k_proj): Linear(in_features=512, out_features=512, bias=False)
                    (v_proj): Linear(in_features=512, out_features=512, bias=False)
                    (o_proj): Linear(in_features=2048, out_features=512, bias=False)
                )
                (post_attention_layernorm): Idefics2RMSNorm((512,), eps=1e-06)
                (pre_ff_layernorm): Idefics2RMSNorm((512,), eps=1e-06)
                (post_ff_layernorm): Idefics2RMSNorm((512,), eps=1e-06)
                (mlp): Idefics2MLP(
                    (gate_proj): Linear(in_features=512, out_features=2048, bias=False)
                    (up_proj): Linear(in_features=512, out_features=2048, bias=False)
                    (down_proj): Linear(in_features=2048, out_features=512, bias=False)
                    (act_fn): SiLU()
                )
            )
        )
        (layernorm): Idefics2RMSNorm((512,), eps=1e-06)
    )
    (decoder): Idefics2PerceiverResampler(
        (layers): ModuleList(
            (0): Idefics2PerceiverLayer(
                (input_latents_layernorm): Idefics2RMSNorm((512,), eps=1e-06)
                (input_context_layernorm): Idefics2RMSNorm((512,), eps=1e-06)
                (cross_attn): Idefics2PerceiverFlashAttention2(
                    (q_proj): Linear(in_features=512, out_features=2048, bias=False)
                    (k_proj): Linear(in_features=512, out_features=512, bias=False)
                    (v_proj): Linear(in_features=512, out_features=512, bias=False)
                    (o_proj): Linear(in_features=2048, out_features=512, bias=False)
                )
                (post_attention_layernorm): Idefics2RMSNorm((512,), eps=1e-06)
                (pre_ff_layernorm): Idefics2RMSNorm((512,), eps=1e-06)
                (post_ff_layernorm): Idefics2RMSNorm((512,), eps=1e-06)
                (mlp): Idefics2MLP(
                    (gate_proj): Linear(in_features=512, out_features=2048, bias=False)
                    (up_proj): Linear(in_features=512, out_features=2048, bias=False)
                    (down_proj): Linear(in_features=2048, out_features=512, bias=False)
                    (act_fn): SiLU()
                )
            )
        )
        (layernorm): Idefics2RMSNorm((512,), eps=1e-06)
    )
)
```

*Listing 1.* Compact description of the architecture of the Perceiver module.

```
(scaler_A): ParameterDict(  (down_proj): Parameter containing: [torch.cuda.FloatTensor of size
↪   1x26x8x1 (cuda:5)])
(scaler_B): ParameterDict(  (down_proj): Parameter containing: [torch.cuda.FloatTensor of size
↪   1x26x8x1 (cuda:5)])
(head): EinMix('bs n_layers n_modules r d_latent -> bs n_layers n_modules r d_lora', 'n_layers
↪   d_latent d_lora', n_layers=26, d_latent=512, r=8, d_lora=11520)
```

*Listing 2.* Compact description of the hypernetwork's head.

```python
def forward(LLM, hypernet, ctx_ids, ctx_attn_mask, input_ids, input_attn_mask):

    # 1) Encode ctx_ids [n_chunks, seq_len] -> features (Z) [n_chunks, n_layers, seq_len, d]
    features = LLM.forward(ctx_ids, ctx_attn_mask).detach()

    # 2) Hypernet: perceiver
    # [n_chunks, n_layers, r, d_latent]
    emb = hypernet.perceiver(features, ctx_attn_mask)

    # 3) Hypernet: head
    # [n_chunks, n_layers, r, d_in + d_out]
    lora_flat = hypernet.head(emb)

    # 4) Combine across context chunks
    # [n_layers, r * n_chunks, d_in + d_out]
    lora = combine_lora(lora_flat, ctx_ids.shape[0])

    # 5) Apply LoRA to base model layers, then run base model
    apply_lora_to_layers(LLM, lora)
    return LLM.forward(input_ids, input_attn_mask)
```

*Listing 3.* Pseudocode for the forward pass of the internalized base model.

```python
PROMPT_TEMPLATE = (
    "You are a creative and helpful assistant.\n"
    "You are tasked with generating questions for reading comprehension tests.\n"
    "You will be given a context and you need to generate questions and corresponding answers from
    ↪   the given context.\n"
    "The questions should be highly specific to the information provided in the context, not general
    ↪   questions that suit any context.\n"
    "**DO NOT** hallucinate or make up information.\n\n"
    "### Instructions ###\n"
    "Generate questions and corresponding answers from the given context. The questions should be
    ↪   highly specific to the "
    "information provided in the context, not general questions that suit any context.\n\n"
    "### Context ###\n"
    "{context}\n\n\n"
    "### Rules ###\n"
    "Rules to follow when generating the questions:\n"
    "1. The questions must be specific to the given context and fully answerable from information
    ↪   present in the given context.\n"
    "2. Ask questions that are fact-seeking based on the information provided.\n"
    "3. Make sure the questions are clear and unambiguous.\n"
    "4. Phrases like 'based on the provided context', 'according to the context', 'in the context',
    ↪   etc., are **NOT ALLOWED** to appear in "
    "the questions.\n"
    "5. The questions should not overlap. They should be diverse, covering many aspects of the
    ↪   context.\n"
    "6. Do not give away too much information in the questions. For example, ask 'Who is X?' instead
    ↪   of 'Who is X that did Y?' when Y is clear from the context.\n"
    "7. Ignore the text formatting of the context, e.g., bold, italic, underline, etc.\n"
    "8. Ignore typos, spacing, and grammatical errors in the context.\n\n"
    "Rules to follow when generating the answers:\n"
    "1. The answers must use the (implied) information provided in the context.\n"
    "2. Phrases like 'based on the provided context', 'according to the context', 'in the context',
    ↪   etc., are **NOT ALLOWED** to appear in "
    "the answers.\n"
    "3. Do not just copy words from the context. Answer the question in your own words.\n"
    "4. The answers should be detailed and comprehensive. Please include additional specific details
    ↪   from the context.\n\n"
    "Respond with {n_qa_pairs} question-answer pairs.\n"
    "Always use proper grammar and punctuation.\n"
    "Try to use different question forms and styles.\n"
    "Use simple words and make sure that the answers are clear and comprehensive.\n\n"
    "The question-answer pairs should be in the following format:\n"
    "Question 1: {{question_1}}\n"
    "Answer 1: {{answer_1}}\n"
    "Question 2: {{question_2}}\n"
    "Answer 2: {{answer_2}}\n"
    "..."
)
```

*Listing 4.* Query generation prompt (first iteration).

```
PROMPT_TEMPLATE_REPEAT = (
    "You are a creative and helpful assistant.\n"
    "You are tasked with generating questions for reading comprehension tests.\n"
    "You will be given a context and you need to generate questions and corresponding answers from
    ↪   the given context.\n"
    "The questions should be highly specific to the information provided in the context, not general
    ↪   questions that suit any context.\n"
    "**DO NOT** hallucinate or make up information.\n\n"
    "### Instructions ###\n"
    "Generate questions and corresponding answers from the given context. The questions should be
    ↪   highly specific to the "
    "information provided in the context, not general questions that suit any context.\n\n"
    "### Context ###\n"
    "{context}\n\n\n"
    "### Example Question-Answer Pairs ###\n"
    "{qa_pairs}\n\n\n"
    "### Rules ###\n"
    "Rules to follow when generating the questions:\n"
    "1. The questions must be specific to the given context and fully answerable from information
    ↪   present in *or* implied from the given context.\n"
    "2. The questions must *not* be redundant with the example questions-answer pairs provided.\n"
    "3. You should prioritize fact-seeking questions. Consider reversal questions, e.g., asking 'What
    ↪   causes X to happen?' is valid when 'Y causes X' is presented in the context.\n"
    "4. If all the facts in the context are already covered by the provided examples, you must
    ↪   generate *more complicated* questions that require reasoning beyond simple information
    ↪   retrieval.\nThis includes asking about information that can be inferred, requiring
    ↪   synthesizing information from multiple parts of the text, or understanding relationships
    ↪   between concepts, events, or individuals mentioned in the context. For example, if the
    ↪   context says 'The Eiffel Tower was completed in 1889 after 2 years of construction', you can
    ↪   ask 'When did the construction of the Eiffel Tower begin?'. Here's another example: if the
    ↪   context says 'Alice is Bob's mother. Bob is Charlie's Dad', you can ask 'Who is Charlie's
    ↪   grandmother?'.\n"
    "5. Phrases like 'based on the provided context', 'according to the context', 'in the context',
    ↪   etc., are **NOT ALLOWED** to appear in "
    "the questions.\n"
    "6. The questions should not overlap. They should be diverse, covering many aspects of the
    ↪   context.\n"
    "7. Do not give away too much information in the questions. For example, ask 'Who is X?' instead
    ↪   of 'Who is X that did Y?' when Y is clear from the context.\n"
    "8. Ignore the text formatting of the context, e.g., bold, italic, underline, etc.\n"
    "9. Ignore typos, spacing, and grammatical errors in the context.\n\n"
    "Rules to follow when generating the answers:\n"
    "1. The answers must use the (implied) information provided in the context.\n"
    "2. Phrases like 'based on the provided context', 'according to the context', 'in the context',
    ↪   etc., are **NOT ALLOWED** to appear in "
    "the answers.\n"
    "3. Do not just copy words from the context. Answer the question in your own words.\n"
    "4. The answers should be detailed and comprehensive. Please include additional specific details
    ↪   from the context.\n\n"
    "Respond with {n_qa_pairs} question-answer pairs.\n"
    "Always use proper grammar and punctuation.\n"
    "Try to use different question forms and styles.\n"
    "Use simple words and make sure that the answers are clear and comprehensive.\n\n"
    "The question-answer pairs should be in the following format:\n"
    "Question 1: {{question_1}}\n"
    "Answer 1: {{answer_1}}\n"
    "Question 2: {{question_2}}\n"
    "Answer 2: {{answer_2}}\n"
    "..."
)
```

*Listing 5.* Query generation prompt (after first iteration).

```
INTX_TEMPLATES = [
    "Answer the question based on the given passages. Only give me the answer and do not output any
    ↪    other words.\n\nQuestion: {input}",
    "Answer without any explanation.\n\nQuestion: {input}",
    "Based on the provided text, what is the answer to the following question? Provide only the
    ↪    answer.\n\nQuestion: {input}",
    "Extract the answer to the question from the text. Be concise. Do not explain.\n\nQuestion:
    ↪    {input}",
    "What is the answer to this question, based on the context? Respond with the answer
    ↪    only.\n\nQuestion: {input}",
    "Provide a direct answer to the question using the given passages. Do not give any
    ↪    explanation.\n\nQuestion: {input}",
    "Answer the question using only information from the provided text. No extra words.\n\nQuestion:
    ↪    {input}",
    "From the passages, answer the question. Just the answer, please.\n\nQuestion: {input}",
    "Give the answer to the question. Do not include any other text.\n\nQuestion: {input}",
    "The answer to the question is in the text. Find it and state it clearly. No need for
    ↪    explanation.\n\nQuestion: {input}",
    "Concisely answer the question based on the text provided. Don't include any other words. Just
    ↪    the answer.\n\nQuestion: {input}",
    "Read the passages and answer the question with the minimal necessary words.\n\nQuestion:
    ↪    {input}",
    "What is the direct response to the question, according to the text? Avoid
    ↪    explanation.\n\nQuestion: {input}",
    "Please provide only the answer to the question, derived from the text.\n\nQuestion: {input}",
    "Using the provided context, answer the question. Output the answer and nothing else.\n\nQuestion:
    ↪    {input}",
    "Identify the answer in the text and present it without elaboration.\n\nQuestion: {input}",
    "Answer the following question based on the text. Your answer should be brief and to the point.
    ↪    No explanation.\n\nQuestion: {input}",
    "Based on the information given, what is the answer to the question? Only state the
    ↪    answer.\n\nQuestion: {input}",
    "Find the answer to the question in the provided passages and write it down. No
    ↪    explanations.\n\nQuestion: {input}",
    "The question is: {input}. Provide the answer based on the text, and nothing more.",
    "Question: {input}\nAnswer directly based on the text provided. No extra words.",
    "Question: {input}\nPlease provide the answer based on the text. No explanation is needed.",
]
```

*Listing 6.* Instruction templates for augmenting generated queries.

```
SELF_RESPONSE_TEMPLATE = (
    "You are an honest and helpful assistant.\n\n"
    "# Provided Information\n"
    "{context}\n\n---\n\n"
    "# System Instruction\n"
    "- The information provided is up-to-date information and/or the user instruction.\n"
    "- When the provided information is not relevant to the question, ***ignore*** it and answer the
    ↪    question based on your knowledge.\n"
    "- If the provided information is related to the question, incorporate it in your response.\n"
    "- If the provided information is an instruction, follow the instruction carefully.\n"
    "\n---\n\n"
    "# User Input\n"
    "{question}"
)
```

*Listing 7.* Prompt template for generating self-responses.

```
QA_PROMPT = (
    "Answer the following question. Output only the answer and do not output any other
    ↪    words.\n\nQuestion: {input}"
)

QASPER_QA_PROMPT = (
    'Answer the question as concisely as you can, using a single phrase or sentence if possible.\nIf
    ↪    the question cannot be answered based on the information in the article, write
    ↪    "unanswerable".\nIf the question is a yes/no question, answer "yes", "no", or "unanswerable".
    ↪    Do not provide any explanation.\n\nQuestion: {input}'
)
```

*Listing 8.* Evaluation prompts.

